# Determining growth rates from bright-field images of budding cells through identifying overlaps

**Julian MJ Pietsch, Alán F Muñoz[†], Diane-Yayra A Adjavon[†], Iseabail Farquhar, Ivan BN Clark, Peter S Swain***

Centre for Engineering Biology and School of Biological Sciences, University of Edinburgh, Edinburgh, United Kingdom

**Abstract** Much of biochemical regulation ultimately controls growth rate, particularly in microbes. Although time-lapse microscopy visualises cells, determining their growth rates is challenging, particularly for those that divide asymmetrically, like *Saccharomyces cerevisiae*, because cells often overlap in images. Here, we present the Birth Annotator for Budding Yeast (BABY), an algorithm to determine single-cell growth rates from label-free images. Using a convolutional neural network, BABY resolves overlaps through separating cells by size and assigns buds to mothers by identifying bud necks. BABY uses machine learning to track cells and determine lineages and estimates growth rates as the rates of change of volumes. Using BABY and a microfluidic device, we show that bud growth is likely first sizer- then timer-controlled, that the nuclear concentration of Sfp1, a regulator of ribosome biogenesis, varies before the growth rate does, and that growth rate can be used for real-time control. By estimating single-cell growth rates and so fitness, BABY should generate much biological insight.

## Editor's evaluation

The authors develop important machine-learning approaches to extract single-cell growth rates and show convincing evidence that their methods can yield insight into growth control. They also introduce compelling new methodologies for several other aspects of automated image analysis.

***For correspondence:**
peter.swain@ed.ac.uk

[†]These authors contributed equally to this work

**Competing interest:** The authors declare that no competing interests exist.

## Introduction

For microbes, growth rate correlates strongly with fitness (*Orr, 2009*). Cells increase growth rates through balancing their synthesis of ribosomes with their intake of nutrients (*Broach, 2012*; *Levy and Barkai, 2009*; *Scott et al., 2014*) and target a particular size through coordinating growth with division (*Johnston et al., 1977*; *Jorgensen et al., 2004*; *Di Talia et al., 2007*; *Turner et al., 2012*). Metazoans, too, not only coordinate growth over time but also in space to both size and position cells correctly (*Ginzberg et al., 2015*).

To understand how organisms regulate growth rate, studying single cells is often most informative (*Murugan et al., 2021*). Time-lapse microscopy, particularly with microfluidic technology to control the extracellular environment (*Locke and Elowitz, 2009*; *Bennett and Hasty, 2009*), has been pivotal, allowing, for example, studies of the cell-cycle machinery (*Di Talia et al., 2007*), of the control of cell size (*Ferrezuelo et al., 2012*; *Schmoller et al., 2015*; *Soifer et al., 2016*), of antibiotic effects (*Coates et al., 2018*; *El Meouche and Dunlop, 2018*), of the response to stress (*Levy et al., 2012*; *Granados et al., 2017*; *Granados et al., 2018*), of feedback between growth and metabolism (*Kiviet et al., 2014*), and of ageing (*Chen et al., 2017*).

For cells that bud, like *Saccharomyces cerevisiae*, estimating an instantaneous growth rate for individual cells is challenging. *S. cerevisiae* grows by forming a bud that increases in size while the volume of the rest of the cell remains relatively unchanged. Although single-cell growth rate is typically reported as the rate of change of volume (*Ferrezuelo et al., 2012*; *Soifer et al., 2016*; *Chandler-Brown et al., 2017*; *Leitao and Kellogg, 2017*; *Garmendia-Torres et al., 2018*; *Litsios et al., 2019*), which approximates a cell's increase in mass, these estimates rely on solving multiple computational challenges: accurately determining the outlines of cells – particularly buds – in images, extrapolating these outlines to volumes, tracking cells over time, assigning buds to the appropriate mother cells, and identifying budding events. Growth rates for budding yeast are therefore often only reported for isolated cells using low-throughput and semi-automated methods (*Ferrezuelo et al., 2012*; *Litsios et al., 2019*). In contrast, for rod-shaped cells that divide symmetrically, like *Escherichia coli*, the growth rate can be found more simply, as the rate of change of a cell's length (*Kiviet et al., 2014*).

A particular difficulty is identifying cell boundaries because neighbouring cells in images often overlap: like other microbes, yeast grows in colonies. Although samples for microscopy are often prepared to encourage cells to grow in monolayers (*Locke and Elowitz, 2009*), growth can be more complex because cells inevitably have different sizes. We observe substantial and frequent overlaps between buds and neighbouring cells in ALCATRAS microfluidic devices (*Crane et al., 2014*). Inspecting images obtained by others, we believe overlap is a widespread, if undeclared, problem: it occurs during growth in the commercial CellASIC devices (*Wood and Doncic, 2019*; *Dietler et al., 2020*), against an agar substrate (*Falconnet et al., 2011*; *Soifer et al., 2016*), in a microfluidic dissection platform (*Litsios et al., 2019*), and in microfluidic devices requiring cells to be attached to the cover slip (*Hansen et al., 2015*).

Yet only a few algorithms allow for overlaps (*Bakker et al., 2018*; *Lu et al., 2019*) despite software to automatically identify and track cells in bright-field and phase-contrast images being well established (*Gordon et al., 2007*; *Falconnet et al., 2011*; *Versari et al., 2017*; *Bakker et al., 2018*; *Wood and Doncic, 2019*) and enhanced with deep learning (*Falk et al., 2019*; *Lu et al., 2019*; *Dietler et al., 2020*; *Stringer et al., 2021*). For example, the convolutional neural network U-net (*Ronneberger et al., 2015a*), a workhorse in biomedical image processing, identifies which pixels in an image are likely from cells, but researchers must find individual cells from these predictions using additional techniques. Even then different instances of cells typically cannot overlap (*Falk et al., 2019*; *Dietler et al., 2020*). Other deep-learning approaches, like Mask-RCNN (*He et al., 2017*) and extended U-nets like StarDist (*Schmidt et al., 2018*), can identify overlapping instances in principle, but typically do not, either by implementation (*Schmidt et al., 2018*) or by the labelling of the training data (*Lu et al., 2019*). Furthermore, assigning lineages and births is often performed manually (*Ferrezuelo et al., 2012*; *Chandler-Brown et al., 2017*) or through fluorescent markers (*Soifer et al., 2016*; *Garmendia-Torres et al., 2018*; *Cuny et al., 2022*), but such markers require an imaging channel.

Here, we describe the Birth Annotator for Budding Yeast (BABY), a complete pipeline to determine single-cell growth rates from label-free images of budding yeast. In developing BABY, we solved multiple image-processing challenges generated by cells dividing asymmetrically. BABY resolves instances of overlapping cells – buds, particularly small ones, usually overlap with their mothers or neighbours – by extending the U-net architecture with custom training targets and then applying additional image processing. It tracks cells between time points with a machine-learning algorithm, which is able to resolve any large movements of cells from one image to the next, and assigns buds to their mothers, informed by the U-net. These innovations improve performance. BABY produces high-fidelity time series of the volumes of both mother cells and buds and so the instantaneous growth rates of single cells.

Using BABY, we see a peak in growth rate during the S/G2/M phase of the cell cycle and show that this peak indicates where the bud's growth transitions from being sizer- to timer-controlled. Studying Sfp1, an activator of ribosome synthesis, we observe that fluctuations in this regulator's nuclear concentration correlate with but precede those in growth rate. Finally, we demonstrate that BABY enables real-time control, running an experiment where changes in the extracellular medium are triggered automatically when the growth of the imaged cells crosses a pre-determined threshold.

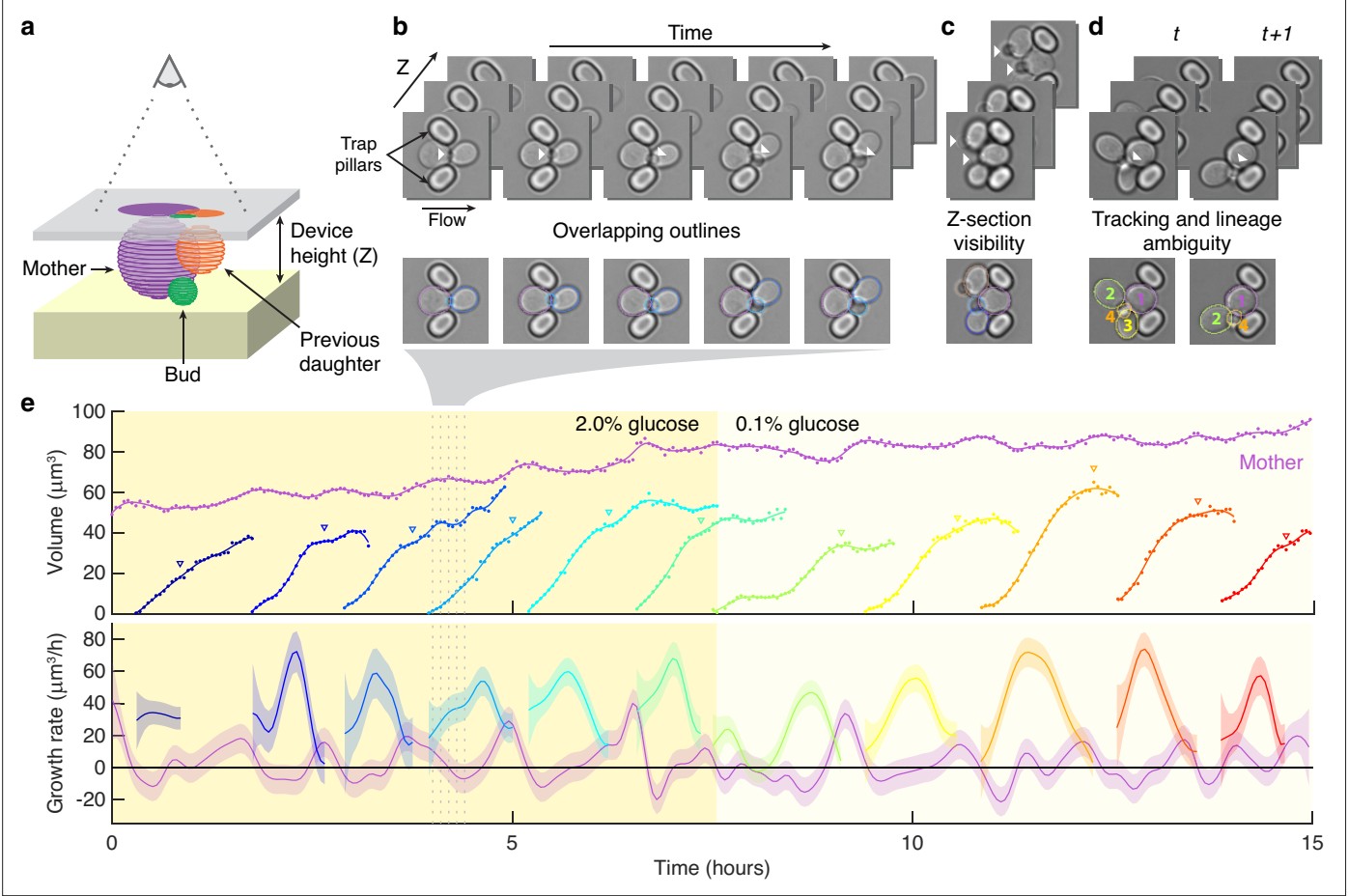

**Figure 1.** Reliably identifying individual cells makes automatically segmenting label-free cells that bud challenging. (**a**) A schematic of a budding cell constrained in a microfluidic device showing how a mother cell can produce a bud beneath the previous daughter. The microscope, denoted by the eye, sees a projection of these cells. (**b**) A time series of bright-field images of budding yeast trapped in an ALCATRAS microfluidic device (*Crane et al., 2014*), in which a growing bud (white arrowheads) overlaps with both its sister and mother. On the duplicated images below, we show outlines produced by BABY. (**c**) Bright-field images of growing buds (white arrowheads) taken at different focal planes demonstrate how the appearance of small buds may change. (**d**) Cells can move substantially from image to image. Here medium flowing through the microfluidic device causes a cell to wash out between time points and the remaining cells to pivot. We indicate the correct lineage assignment by white arrowheads and the correct tracking by the numbers within the BABY outlines. (**e**) We show a time series of a mother (purple) and its buds and daughters for a switch from 2% to 0.1% glucose using volumes and growth rates estimated by BABY. Bud growth rates are truncated to the predicted time of cytokinesis (triangles). Shaded areas are twice the standard deviation of the fitted Gaussian process.

The online version of this article includes the following figure supplement(s) for figure 1:

**Figure supplement 1.** Growth rates are highest for small buds.

## Results

### Segmenting overlapping cells using a multi-target convolutional neural network

To estimate single-cell growth rates from time-lapse microscopy images, correctly identifying cells is essential. Poorly defined outlines, missed time points, and mistakenly joined cells all degrade accuracy.

Segmenting asymmetrically dividing cells, such as budding yeast, is challenging. The differing sizes of the mothers and buds makes each appear and behave distinctly, yet identifying buds is crucial because they have the fastest growth rates (*Ferrezuelo et al., 2012*; *Figure 1—figure supplement 1*). Even when constrained in a microfluidic device, buds imaged in a single Z section often appear to overlap with their mother and neighbouring cells (*Figure 1a and b*). If an algorithm is able to separate the cells, the area of either the bud or the neighbouring cells is often underestimated, and the bud may even be missed entirely. Buds also move more in the Z-axis relative to mother cells, changing how

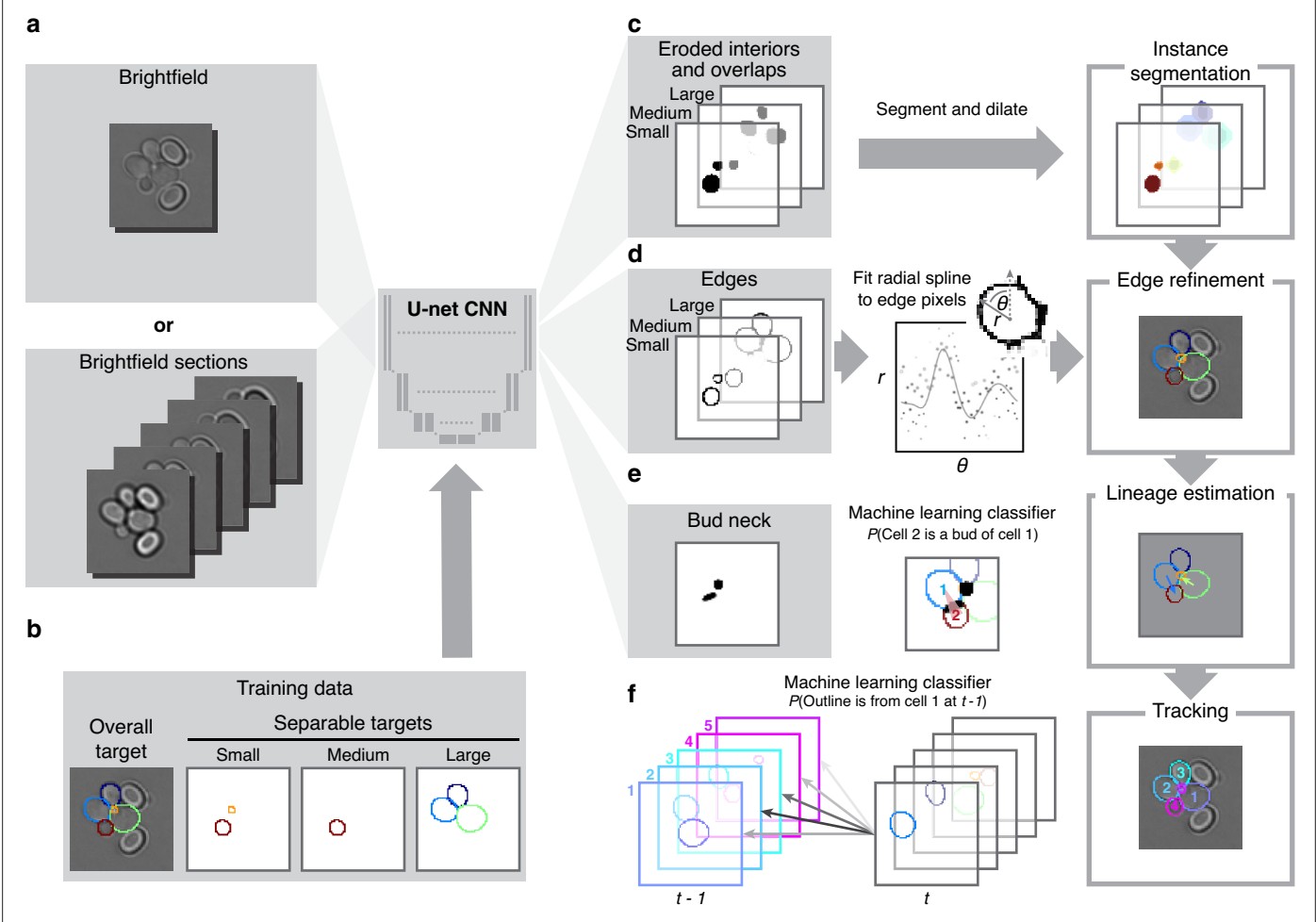

**Figure 2.** BABY uses multiple bright-field Z-sections, a multi-target convolutional neural network followed by a custom segmentation algorithm, and two machine-learning classifiers to identify cells and their buds reliably from image to image. (**a**) Either single or multiple, we typically use five, bright-field Z-sections are input into a multi-target U-net CNN. (**b**) The curated training data comprises multiple outlines that we categorise by size to reduce overlaps between cells within each category. (**c**) We train the CNN to predict a morphological erosion of the target cell images, which act as seeds for segmenting instances of cells. (**d**) We use edge targets from the CNN to refine each cell's outline, parameterised as a radial spline. (**e**) We use a bud-neck target from the CNN and metrics characterising the cells' morphologies to estimate the probability that a pair of cells is a mother and bud via a machine-learning classifier. (**f**) Another classifier uses the same morphological metrics to estimate the probability that an outline in the previous time point matches the current one.

they appear in bright-field images (*Figure 1c*). Depending on the focal plane, a bud may be difficult to detect by eye. Nevertheless, our BABY algorithm maintains high reliability (*Figure 1e*).

Like others, we use a U-net, a convolutional neural network (CNN) with an architecture that aims to balance accuracy with simplicity (*Ronneberger et al., 2015a*), and our main innovation is in the choice of training targets. We improve performance further by using multiple Z-sections (*Figure 2a*, *Figure 3—figure supplement 1*), although BABY can predict overlapping outlines from a single 2D image, and we train on single images.

Inspecting cells, we noted that how much and how often they overlap depends on their size (*Appendix 1—figure 2*). Most overlaps occur between mid-sized cells and buds with sizes in the range expected for fast growth (*Figure 1—figure supplement 1a*). We therefore divided our training data into three categories based on cell size. From each annotated image – a single Z section, we generated up to three new training images: one showing any cells in the annotated image in our small category, one showing any in the medium category, and one for any large cells. We decreased any remaining overlaps in these training images by applying a morphological erosion (*Figure 2b*; *Appendix 1—figures 1 and 2*), shrinking the cells by removing pixels from their boundaries. Although

this transformation does reduce the number of overlapping cells, it may undermine accuracy when we segment the cells. We therefore include the boundary pixels of all the cells in the original annotated image (*Figure 2d*) as a training target. To complement this size-based approach, we add another training target: the overlaps between any pair of cells irrespective of their size in the annotated image.

A final target is the 'bud neck' (*Figure 2e*), which helps to identify which bud belongs to which cell. In bright-field images, cytokinesis is sometimes visible as a darkening of the bud neck, indicating that these images contain information on cytokinesis that the U-net can potentially learn. We manually created the training data to avoid ambiguity, annotating bright-field images and then generating binary ones showing only bud necks.

The targets of the U-net therefore comprise the cell interiors and boundaries, separated by size, all overlaps between cells, and the bud necks. Using a four-layer U-net, we achieved high accuracy for predicting the cell interiors early in training and with around 600 training images (1,813 annotated cells in total; *Figure 2c* & *Appendix 1—figure 3c*). The performance on bud necks is lower (*Appendix 1—figure 3e*), but sufficient because we supplement this information with morphological features when assigning buds. Unlike others (*Lugagne et al., 2018*; *Bakker et al., 2018*), we do not need to explicitly ignore objects in the image because the network learns to disregard both the traps in ALCATRAS devices and any debris.

To determine smooth cell boundaries, we apply additional image processing to the U-net's outputs. First, we reverse the morphological erosion that we applied to the training data (*Appendix 1—figure 4*), adding pixels to the U-net's predicted cell interiors. Second, and like the StarDist (*Schmidt et al., 2018*) and DISCO algorithms (*Bakker et al., 2018*), we parameterise the cell boundaries using a radial representation because we expect yeast cells to appear elliptical – although we can describe any star-convex shape. We fit radial splines with 4–8 rays depending on the cell's size to a re-weighted version of its boundary pixels predicted by the U-net (*Appendix 1—figure 5*). On test images, the resulting cell boundaries improve accuracy compared to using the U-net's predictions directly (*Figure 3—figure supplement 1*).

Other features further improve performance. We developed a graphical user interface (GUI) to label and annotate overlapping cells (*Appendix 4—figure 1*). With the GUI, we create a 2D binary image of each cell's outline by using all Z sections together to annotate the outline from the Z section where the cell is most in focus. We also wrote scripts to optimise BABY's hyper-parameters during training (Methods).

We find that BABY outperforms alternatives (*Figure 3a*), even when we retrain these alternatives with the BABY training data. For larger cell sizes, BABY performs comparably with two algorithms based on deep learning: Cellpose (*Stringer et al., 2021*; *Pachitariu and Stringer, 2022*), a generalist algorithm, and YeaZ (*Dietler et al., 2020*), an algorithm optimised for yeast. For smaller cell sizes, BABY performs better, identifying buds overlapping with mother cells that both Cellpose and YeaZ miss (*Figure 3b*). To assess its generality, we turned to time-lapse images of yeast microcolonies, training a BABY model on only 6% of the annotated microcolony training data provided by YeaZ and evaluating its performance on the remaining images. BABY performs competitively (*Figure 3—figure supplement 2*), and even detects buds that were neither annotated in the ground truth nor detected by Cellpose and YeaZ (*Figure 3—figure supplement 3*).

## Using machine learning to track lineages robustly

To determine growth rates, we should estimate both the mother's and the bud's volumes because most growth occurs in the bud (*Hartwell and Unger, 1977*; *Ferrezuelo et al., 2012*). We should therefore track cells from one time point to the next and correctly identify, track, and assign buds to their mothers (*Appendix 2—figure 1*).

This last task of assigning a bud to its mother is challenging (*Figure 1d*). Buds frequently first appear surrounded by cells, displacing their neighbours as they grow (*Figure 1b*), obfuscating which is the mother. Both mother and bud can react to the flow of medium: buds often pivot around their mother, with other cells sometimes moving too (*Figure 1d*). If tracked incorrectly, a pivoting bud may be misidentified as a new one.

By combining the U-net's predicted bud-necks with information on the shape of the cells, we accurately assign buds. Our approach is first to identify cells in an image that are likely buds and then to assign their mothers. We use a standard classification algorithm to estimate the probability that each

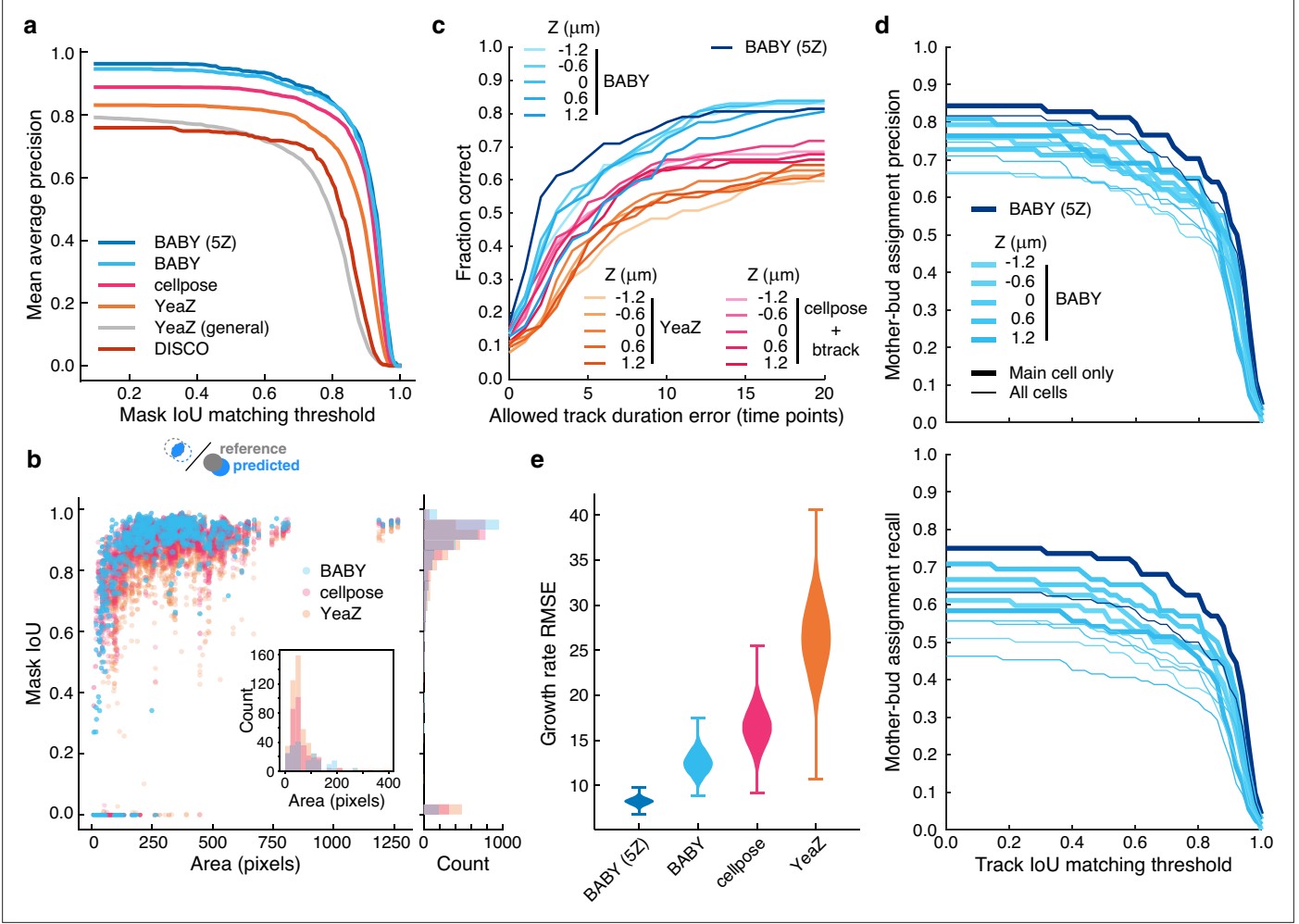

**Figure 3.** BABY outperforms other algorithms for segmenting, tracking, and particularly for estimating growth rates. (**a**) Comparing the intersection-over-union (IoU) score (Methods) between manually curated single cells and those predicted by the BABY, Cellpose, YeaZ, and DISCO algorithms shows that BABY performs best, particularly with five Z sections as input (5Z). We show the performance of the generalist YeaZ model and the Cellpose and YeaZ algorithms retrained on the BABY training data. (**b**) BABY performs particularly well for smaller cell sizes. Inset: counts of curated cells missed by each algorithm. (**c**) BABY finds a higher fraction of complete tracks than either YeaZ or Cellpose, an algorithm only for segmentation and trained on BABY data, combined with btrack (*Ulicna et al., 2021*), a tracking algorithm. We show the results for each Z section separately because BABY is the only algorithm that can use more than one. (**d**) We show BABY's precision and recall for correctly assigning mother and bud tracks in the tracking evaluation data set as a function of the threshold for defining matching tracks. Performance is best for the central trapped cell. We are unaware of any other algorithms performing mother-bud assignment directly from bright-field images with which to compare. (**e**) By accurately detecting and estimating buds with small volumes, BABY also shows the smallest Root Mean Squared Error (RMSE) when comparing predicted bud growth rates with those derived from a manually curated set of time series of randomly selected mother-bud pairs from four different growth conditions. To highlight the importance of segmentation quality for estimating growth rates, we matched outlines to ground truth ignoring any tracking errors. We used $10^4$ bootstraps of 90% of the ground truth data (209 estimates of growth rate from 9 buds) to find the distributions of RMSE.

The online version of this article includes the following figure supplement(s) for figure 3:

**Figure supplement 1.** Optimising edges and using multiple Z sections as inputs improves segmentation.

**Figure supplement 2.** BABY is competitive with existing algorithms for segmenting microcolonies.

**Figure supplement 3.** BABY detects buds that were missing in the YeaZ training images.

**Figure supplement 4.** BABY produces fewer missing tracks than existing other algorithms.

**Figure supplement 5.** All algorithms tested track similarly when assessed with a generalised metric.

**Figure supplement 6.** A peak in the bud's growth rate predicts cytokinesis.

pair of cells in an image are a mother and bud (*Appendix 2—figure 4*). This classifier uses as inputs both the predicted bud-necks and the cells' morphological characteristics, which we extract from the segmented image – one with every cell identified. For each bud, we assign its mother using information from both the current image and the past: the mother is the cell with the highest accumulated probability of pairing with the bud over all previous images showing both cells (Appendix 2).

We use another classifier-based approach for tracking. The classifier estimates the probability that each pair of cells in two segmented images at different time points, with one cell in the first image and the other in the second, are the same cell (*Figure 2f*). To be able to track cells that pivot (*Figure 1d*), we train two classifiers: the first using only the cells' morphological characteristics and the second using these characteristics augmented with the distance between the cells, a more typical approach (*Falconnet et al., 2011*; *Bakker et al., 2018*; *Garmendia-Torres et al., 2018*; *Wood and Doncic, 2019*; *Dietler et al., 2020*) but one that often misses pivoted cells. If the results of the first classifier are ambiguous, we defer to the second (Appendix 2). We aggregate tracking predictions over the previous three time points to be robust to transient errors in image processing and in imaging, like a loss of focus. Our algorithm also identifies unmatched cells, which we treat either as new buds or cells moved by the flow of medium: cells may disappear from one time point to the next or be swept downstream and appear by a trap.

BABY finds more complete or near-complete tracks than other algorithms (*Figure 3c*, *Figure 3—figure supplement 4*). Cellpose does not perform tracking, and we therefore used the btrack algorithm (*Ulicna et al., 2021*) to track outlines segmented by a Cellpose model trained on the BABY training data. We assessed each algorithm against manually curated data by calculating the intersection-over-union score (IoU) between cells in a ground-truth track with those in a predicted track. We report both the fraction of ground-truth tracks that a predicted track matches, to within some tolerance for missing time points (*Figure 3c*), and the track IoU – the number of time points where the cells match relative to the total duration of both tracks (*Figure 3—figure supplement 4*). If multiple predicted tracks match a ground-truth track, we use the match with the highest track IoU, and any predicted tracks left unassigned have a track IoU of zero. BABY excels because it detects buds early, which both increases the track IoU and prevents new buds being tracked to an incorrect cell.

We also compared tracking performance using a more general metric, the Multiple Object Tracking Accuracy (MOTA) (*Bernardin et al., 2006*; *Figure 3—figure supplement 5*). With this metric, all methods performed similarly, though Cellpose with btrack appeared more robust to the given Z section. The MOTA score is ideal when there are numerous objects to track and frequent mismatches. Accurately measuring the duration of tracks is necessary to report division times, and so our metrics penalise track splitting, where a ground-truth track is erroneously split into two predicted tracks. The penalty for a single tracking error can therefore differ depending on when that error happens. In contrast, MOTA explicitly avoids penalising splitting errors.

We are unaware of other algorithms that assign buds to mothers using only bright-field images and so report only BABY's precision and recall for correctly pairing mother and bud tracks on the manually curated data set (*Figure 3d*). Microfluidic devices with traps typically capture one central cell per trap, so we present both the performance for all cells and for only these central cells. BABY requires a mother and bud to be paired over at least three time points (15 min or an eighth of a cell-cycle in 2% glucose), and so when considering all cells, BABY fails to recall multiple mother-bud pairs because daughters of the central cell are often washed away soon after producing a bud.

## Estimating growth rates

From the time series of segmented cells, we estimate instantaneous single-cell growth rates as time derivatives of volumes (Appendix 3). We independently estimate the growth rates of mothers and buds, each from their own time series of volumes. A cell's growth rate, the rate of change of the total volume of a mother and bud, is their sum. To find a cell's volume from its segmented outline, we use a conical method (*Gordon et al., 2007*; *Figure 1e*) and make only weak assumptions to find growth rates from these volumes. Researchers have modelled single-cell growth rates in yeast as bilinear (*Cookson et al., 2010*; *Ferrezuelo et al., 2012*; *Leitao and Kellogg, 2017*; *Garmendia-Torres et al., 2018*) and exponential (*Di Talia et al., 2007*; *Godin et al., 2010*; *Soifer et al., 2016*; *Chandler-Brown et al., 2017*), but that choice has implications for size control (*Turner et al., 2012*). Instead, we use a Gaussian process to both smooth the time series of volumes and to estimate their time derivatives

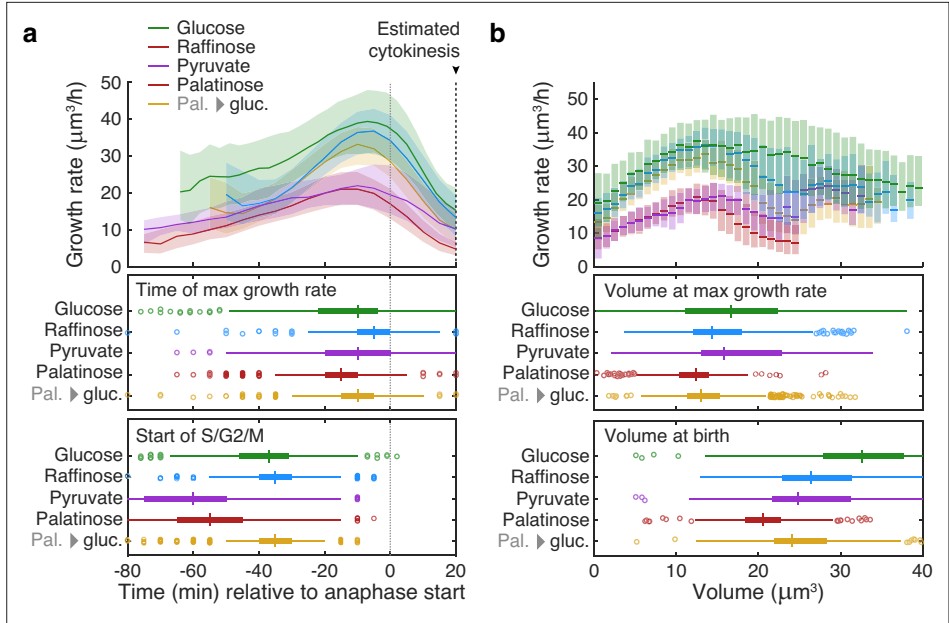

**Figure 4.** Buds reach similar sizes as their growth rate peaks regardless of carbon source. (**a**) Although buds grow faster in richer media, the time of the maximal growth rate relative to the start of anaphase is approximately constant, unlike the duration of the mothers' S/G2/M phases. We grew cells in 2% glucose (data for 1014 cell cycles), 2% raffinose (803 cycles), 2% pyruvate (270 cycles), 2% palatinose (393 cycles), or in 2% glucose after a switch from palatinose (pal. → gluc.; 842 cycles). We show median bud growth rates with the interquartile range shaded and estimate the timing of anaphase from a fluorescently tagged nuclear marker (Nhp6A-mCherry; Appendix 6) and the start of S phase by when a bud first appears. (**b**) Binning median bud growth rates according to volume, with the interquartile range shaded, shows that the bud volumes when their growth rate is maximal are more similar in all carbon sources than those at birth, taken as 20 min after start of anaphase (*Leitao and Kellogg, 2017*).

The online version of this article includes the following figure supplement(s) for figure 4:

**Figure supplement 1.** Growth rates estimated with BABY show expected correlations with volume.

---

(*Swain et al., 2016*), and so make assumptions only on the class of functions that describe growth rather than choosing a particular functional form. Like others (*Cookson et al., 2010*; *Ferrezuelo et al., 2012*), we observe periodic changes in growth rate across the cell cycle (*Figure 1e*).

BABY estimates growth rates more reliably than other algorithms (*Figure 3e*). We manually curated time series of randomly selected mother-bud pairs from four different growth conditions, annotating both mother and bud from the bud's first appearance to the appearance of the next one (436 outlines total). BABY best reproduces the growth rates derived from this ground truth.

## BABY provides new insights and experimental designs

### Nutrient modulation of birth size occurs after the peak in growth rate

Using a fluorescent marker for cytokinesis (*Figure 3—figure supplement 6a*), we observed that cellular growth has two phases (*Figure 3—figure supplement 6b–c*). During G1, the mother's growth rate peaks; during S/G2/M, which we identify by the cells having buds, the bud dominates growth with its growth rate peaking approximately midway to cytokinesis (*Ferrezuelo et al., 2012*).

This tight coordination between bud growth rate and cytokinesis suggested that the peak in bud growth rate preceding cytokinesis may mark a regulatory transition. Comparing growth rates over S/G2/M for buds in different carbon sources, we found that the maximal growth rate occurs at similar times relative to cytokinesis despite substantial differences in the duration of the S/G2/M phases (*Figure 4a*).

Daughters born in rich media are larger than those born in poor media, and some of this regulation occurs during S/G2/M (*Johnston et al., 1977*; *Jorgensen et al., 2004*; *Leitao and Kellogg, 2017*). Understanding the mechanism, however, is confounded by the longer S/G2/M phases in poorer

media (*Leitao and Kellogg, 2017*; *Figure 4a*), which counterintuitively allow daughters that should be smaller longer to grow.

Given that the time between maximal growth and anaphase appears approximately constant in different carbon sources (*Figure 4a*), we hypothesised that the growth rate falls because the bud has reached a critical size. Compared to how their sizes vary immediately after cytokinesis, buds have similar sizes when their growth rates peak — in all carbon sources (*Figure 4b*): the longer S/G2/M phase in poorer media compensates the slower growth rates. During the subsequent constant time to cytokinesis, the faster growth in richer carbon sources would then generate larger daughters, and we observe that the bud's average growth rate correlates positively with the volume of the daughter it becomes (*Figure 4—figure supplement 1*). Cells likely therefore implement some size regulation in S/G2/M as they approach cytokinesis.

Although such regulation in M phase is known (*Leitao and Kellogg, 2017*; *Garmendia-Torres et al., 2018*), our data suggest a sequential mechanism to match size to growth rate, with a nutrient-independent sizer followed by a nutrient-dependent timer. To detect the peak in bud growth generated by the sizer, cells may use Gin4-related kinases (*Jasani et al., 2020*).

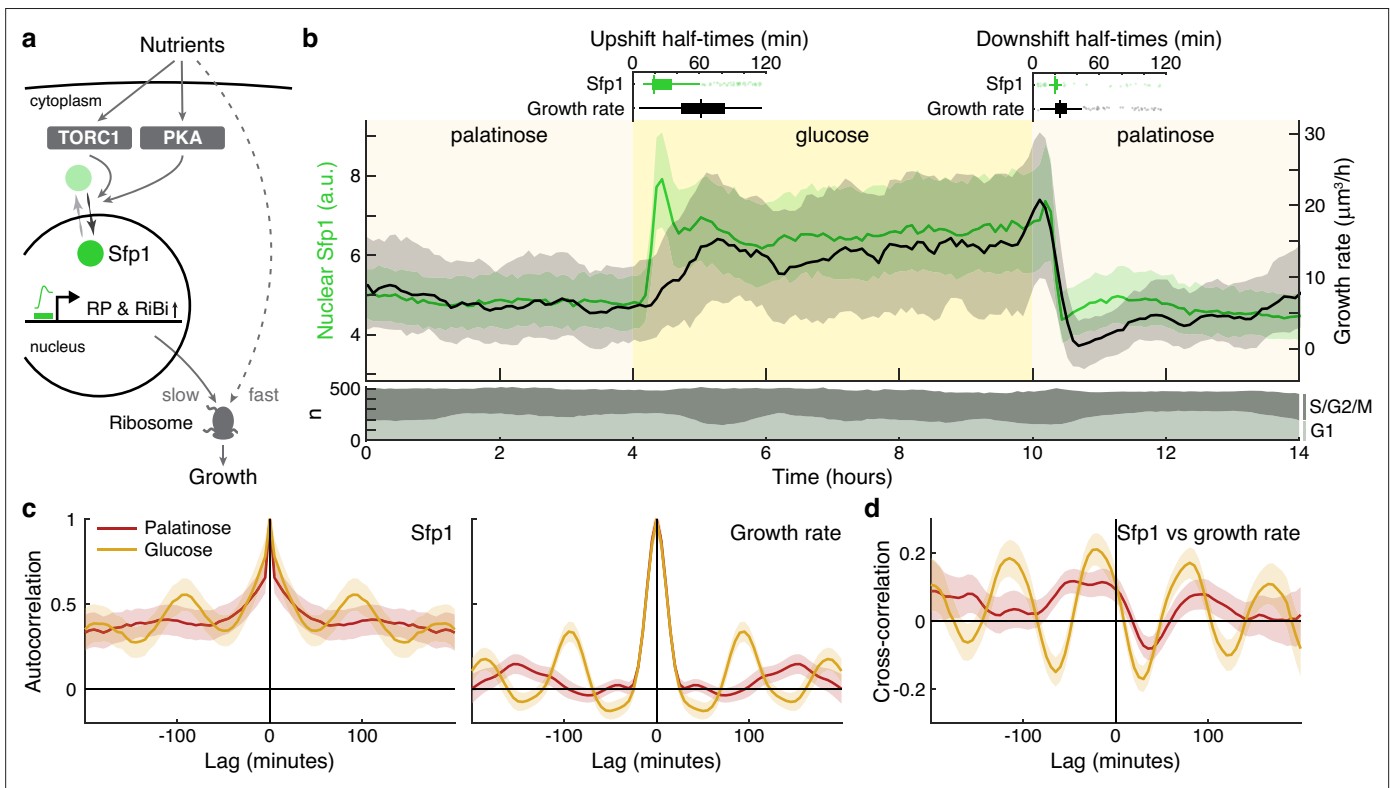

**Figure 5.** The translocation dynamics of the ribosomal regulator Sfp1 anticipate changes in single-cell growth rates. (**a**) The transcription factor Sfp1 is phosphorylated by TORC1 and likely PKA when extracellular nutrients increase and moves into the nucleus, where it promotes synthesis of ribosomes and so higher growth rates. (**b**) Growth rate follows changes in Sfp1's nuclear localisation if nutrients decrease but lags if nutrients increase. We show the median time series of Sfp1-GFP localised to the nuclei of mother cells (green) and the summed bud and mother growth rates (black) for cells switched from 2% palatinose to 2% glucose and back. Shading shows interquartile ranges. We filtered data to those cell cycles that could be unambiguously split into G1 and S/G2/M phases by a nuclear marker, and we display the number in each phase in the lower plot. Above the switches of media, we show box plots for the distributions of single-cell half-times: the time of crossing midway between each cell's minimal and maximal values. (**c**) The mean single-cell autocorrelation of nuclear Sfp1 and the summed mother and bud growth rates are periodic because both vary during the cell cycle. We calculate the autocorrelations for constant medium using data four hours before each switch (Appendix 7). Shading shows the 95% confidence interval. (**d**) The mean cross-correlation between nuclear Sfp1 and the summed mother and bud growth rate shows that fluctuations in Sfp1 precede those in growth, with the correlation peaking at negative lags.

The online version of this article includes the following figure supplement(s) for figure 5:

**Figure supplement 1.** Irrespective of cell cycle phase, growth rates transiently drop for a shift to a poorer carbon source.

## Changes in ribosome biogenesis precede changes in growth

An important advantage of the BABY algorithm is that we can estimate single-cell growth rates without fluorescence markers, freeing fluorescence channels for other reporters. Here we focus on Sfp1, a transcription factor that helps coordinate ribosome synthesis with the availability of nutrients (*Jorgensen et al., 2004*).

Sfp1 promotes the synthesis of ribosomes by activating the ribosomal protein (RP) and ribosome biogenesis (RiBi) genes (*Jorgensen et al., 2004*; *Albert et al., 2019*). Upon being phosphorylated directly by TORC1 and likely protein kinase A (*Jorgensen et al., 2004*; *Lempiäinen et al., 2009*; *Singh and Tyers, 2009*) – two conserved nutrient-sensing kinases, Sfp1 enters the nucleus (*Figure 5a*). In steady-state conditions, levels of ribosomes positively correlate with growth rate (*Metzl-Raz et al., 2017*), and we therefore assessed whether Sfp1's nuclear localisation predicts changes in instantaneous single-cell growth rates.

Shifting cells from glucose to the poorer carbon source palatinose and back again, we observed that Sfp1 responds quickly to both the up- and downshifts and that growth rate responds as quickly to downshifts, but more slowly to upshifts (*Figure 5b*). As a target of TORC1 and PKA, Sfp1 acts as a fast read-out of the cell's sensing of a change in nutrients (*Granados et al., 2018*). In contrast, synthesising more ribosomes is likely to be slower and explains the lag in growth rate after the upshift. The fast drop in growth rate in downshifts is more consistent, however, with cells deactivating ribosomes, rather than regulating their numbers. Measuring the half-times of these responses (*Figure 5b* boxplots), there is a mean delay of 30 ± 2 minutes (95% confidence; $n = 245$) from Sfp1 localising in the nucleus to the rise in growth rate in the upshift. This delay is only 8 ± 1 minutes (95% confidence; $n = 336$) in the downshift, and downshift half-times are less variable than those for upshifts, consistent with fast post-translational regulation. Although changes in Sfp1 consistently precede those in growth rate, the higher variability in half-times for the growth rate is not explained by Sfp1's half-time (Pearson correlation 0.03, $p = 0.6$).

By enabling both single-cell fluorescence and growth rates to be measured, BABY permits correlation analyses (*Kiviet et al., 2014*; Appendix 7). Both Sfp1's activity and the growth rate vary during the cell cycle. The autocorrelation functions for nuclear Sfp1 and for the growth rate are periodic with periods consistent with cell-division times (*Figure 5c*): around 90 min in glucose and 140 min in palatinose for Sfp1; and 95 min and 150 min for the growth rate. If Sfp1 acts upstream of growth rate, then its fluctuations in nuclear localisation should precede fluctuations in growth rate. Cross-correlating nuclear Sfp1 with growth rate shows that fluctuations in Sfp1 do lead those in growth rate, by an average of 25 min in glucose and by 50 min in palatinose (*Figure 5d*). Nevertheless, the weak strength of this correlation suggests substantial control besides Sfp1.

During the downshift, we note that the growth rate transiently drops to zero (*Figure 5b*), irrespective of a cell's stage in the cell cycle (*Figure 5—figure supplement 1*), and there is a coincident rise in the fraction of cells in G1 (*Figure 5b* bottom), suggesting that cells arrest in that phase.

## Using growth rate for real-time control

With BABY, we can use growth rate as a control variable in real time because BABY's speed and accuracy enables cells to be identified in images and their growth rates estimated during an experiment (*Figure 6a*). As an example, we switched the medium to a poorer carbon source and used BABY to determine how long to keep cells in this medium if we want 50% to have resumed dividing before switching back to the richer medium (Appendix 8). After 5 hr in glucose, we switched the carbon source to ethanol, or galactose – *Figure 6—figure supplement 1*. There is a lag in growth as cells adapt. Using BABY, we automatically determined the fraction of cells that have escaped the lag at each time point — those cells that have at least one bud or daughter whose growth rate exceeds a threshold (*Figure 6b*). The software running the microscopes reads this statistic and triggers the switch back to glucose when 50% of the cells have escaped (*Figure 6c*). We note that all cells resume dividing in glucose and initially grow synchronously because of the rapid change of media. This synchrony is most obvious in those cells that did not divide in ethanol (*Figure 6c*).

This proof-of-principle shows that BABY is applicable for more complex feedback control, where a desired response is achieved by comparing behaviour with a computational model to predict the necessary inputs, such as changes in media (*Harrigan et al., 2018*; *Milias-Argeitis et al., 2011*; *Toettcher et al., 2011*; *Uhlendorf et al., 2012*; *Lugagne et al., 2017*; *Menolascina et al., 2014*).

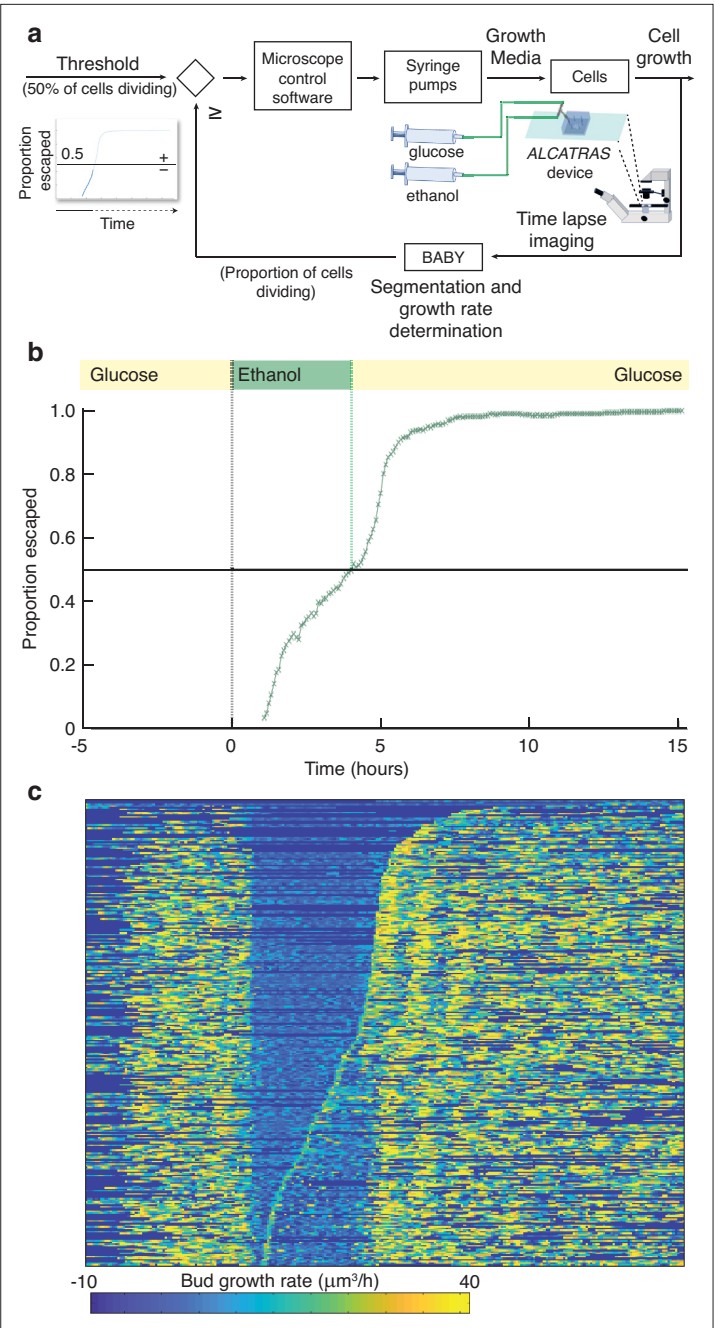

**Figure 6.** BABY allows growth rate to be used as a variable for real-time control. (**a**) By running BABY in real time during a microscopy experiment, we are able to use the cells' growth rate to control changes in media. Following 5 hr in 0.5% glucose, we switch the extracellular medium to one containing 2% ethanol, a poorer carbon source, and cells arrest growth. The images collected are analysed by BABY to determine growth rates. When the majority of cells have resumed dividing, detected by the growth rate of at least one of their buds or daughters exceeding $15 \mu m^3/hr$, the microscopy software triggers a change in pumping and returns glucose to the microfluidic device. (**b**) The fraction of cells that have escaped the lag and resumed dividing increases with the amount of time in ethanol. All cells divide shortly after glucose returns. (**c**) The growth rates of the buds for each mother cell drop in ethanol and resume in glucose. Each row shows data from a single mother cell with the bud growth rate indicated by the heat map. We sort rows by the time each cell resumes dividing in ethanol, with the bottom rows showing the 50% that re-initiated growth.

The online version of this article includes the following figure supplement(s) for figure 6:

**Figure supplement 1.** Changing experimental protocols in real time using growth rates.

Unlike previous approaches though, which typically measure fluorescence, BABY not only allows single-cell fluorescence but also growth rates to be control variables, and growth rate correlates strongly with fitness (*Orr, 2009*).

## Discussion

Here, we present BABY, an algorithm to extract growth rates from label-free, time-lapse images through reliably estimating time series of cellular volumes. We introduce both a segmentation algorithm that identifies individual cells in images even if they overlap and general machine-learning methods to track and assign lineages robustly. The novel training targets for CNNs that we propose, particularly splitting one training image into multiple with each comprising cells of a particular size, should be beneficial not only for other yeasts but for other cell types.

Although BABY detects buds shortly after they form, we stop following a bud as soon as the mother buds again and instead follow the new one. Ideally we would like to identify from bright-field images when a bud becomes an independent daughter cell. We would then know when a mother cell exits M phase and be able to identify their G1 and the (budded) S/G2/M phases. We have partly achieved this task with an algorithm that predicts the end of the peak in the bud's growth rate (Appendix 6), which often occurs at cytokinesis (*Figure 3—figure supplement 6a*; *Appendix 6—figure 1a*). It assigns to within two time points over 60% of the cytokinesis events identified independently using a fluorescent reporter (*Figure 3—figure supplement 6d–e*), but higher accuracy likely needs more advanced techniques.

Indeed, we believe that integrating BABY with other algorithms will improve its performance even further. How Cellpose defines training targets for its CNN appears particularly powerful (*Stringer et al., 2021*; *Pachitariu and Stringer, 2022*), and this formulation could be combined with BABY's size-dependent categorisation. Similarly, for assigning lineages, there are now methods that use image classification to identify division and budding times for cells in traps (*Aspert et al., 2022*), and for tracking, our machine learning approach would benefit from Fourier transforming the images we use, which provides a rich source of features (*Cuny et al., 2022*).

Cell biologists often wish to understand how cells respond to change (*Murugan et al., 2021*), and watching individual cells in real time as their environment alters gives unique insights (*Locke and Elowitz, 2009*). Together time-lapse microscopy, microfluidic technology, and fluorescent proteins allow us to control extracellular environments, impose dynamic changes, and phenotype cellular responses over time. With BABY, we add the ability – using only bright-field images – to measure what is often our best estimate of fitness, single-cell growth rates. The strategies used by cells in their decision making are of high interest (*Perkins and Swain, 2009*; *Balázsi et al., 2011*). With BABY, or comparable software, we are able not only to use fitness to rank each cell's decision-making strategy, but also to investigate the strategies used to regulate fitness itself, through how cells control their growth, size, and divisions.

## Methods

### Strains and media

Strains included in the curated training images were all derivatives of BY4741 (*Brachmann et al., 1998*). We derived both BY4741 Myo1-GFP Whi5-mCherry and BY4741 Sfp1-GFP Nhp6A-mCherry from the respective parent in the *Saccharomyces cerevisiae* GFP collection *Huh et al., 2003* by PCR-based genomic integration of mCherry-Kan $R$ from pBS34 (EUROSCARF) to tag either Whi5 or the chromatin-associated Nhp6A protein. We validated all tags by sequencing. The media used for propagation and growth was standard synthetic complete (SC) medium supplemented either with 2% glucose, 2% palatinose, or 0.5% glucose depending on the starting condition in the microfluidic devices. Cells were grown at 30 °C.

### Microscopy and microfluidics

#### Device preparation and imaging

We inoculated overnight cultures with low cell numbers so that they would reach mid-log phase in 13–16 hr. We diluted cells in fresh medium to $OD_{600}$ of 0.1 and incubated for an additional 3–4 hr

before loading them into microfluidic devices at ODs of 0.3–0.4. To expose multiple strains to the same environmental conditions and to optimise data acquisition, we use multi-chamber versions of ALCATRAS (*Crane et al., 2014*; *Granados et al., 2017*; *Crane et al., 2019*), which allow for either three or five different strains to be observed in separate chambers while being exposed to the same extracellular medium. The ALCATRAS chambers were pre-filled with growth medium with added 0.05% bovine serum albumin (BSA) to facilitate cell loading and reduce clumping. We passed all microfluidics media through 0.2 μm filters before use.

We captured images on a Nikon Ti-E microscope using a 60×, 1.4 NA oil immersion objective (Nikon), OptoLED light source (Cairn Research) and sCMOS (Prime95B), or EMCCD (Evolve) cameras (both Photometrics) controlled through custom MATLAB software using Micro-manager (*Edelstein et al., 2014*). We acquired bright-field and fluorescence images at five Z sections spaced 0.6 μm apart. A custom-made incubation chamber (Okolabs) maintained the microscope and syringe pumps containing media at 30 °C.

## Changing the extracellular environment

For experiments in which the cells experience a change of media, two syringes (BD Emerald, 10 ml) mounted in syringe pumps (Aladdin NE-1002X, National Instruments) connected via PTFE tubing (Smiths Medical) to a sterile metal T-junction delivered media through the T-junction and via PTFE tubing to the microfluidic device. Initially the syringe with the first medium infused at 4 μL/min while the second pump was off. To remove back pressure and achieve a rapid switch, we infused medium at 150 μL/min for 40 s from the second pump while the first withdrew at the same rate. The second pump was then set to infuse at 4 μL/min and the first switched off. We reversed this sequence to achieve a second switch in some experiments. Custom Matlab software, via RS232 serial ports, controlled the flow rates and directions of the pumps.

## Birth Annotator for Budding Yeast (BABY) algorithm

The BABY algorithm takes either a stack of bright-field images or a single Z-section as input and coordinates multiple machine-learning models to output individual cell masks annotated for both tracking and lineage relationships.

Central to segmenting and annotating lineages is a multi-target CNN (Appendix 1). Each target is semantic – pixels have binary labels. We define these targets for particular categories of cell size and mask pre-processing steps, chosen to ease both segmenting overlapping instances and assigning lineages. We first identify cell instances as semantic masks and then refine their edges using a radial spline representation.

To track cells and lineages, we use machine-learning classifiers both to link cell outlines from one time point to the next and to identify mother-bud relationships. The classifier converts a feature vector, representing quantitatively how two cell masks are related, into probabilities for two possible classes. For cell tracking, this probability is the probability that the two cells at different time points are the same cell. For assigning lineages, the probability is the probability that the two cells have a mother-bud relationship. We aggregate over time a target of the CNN dedicated to assigning lineages to determine this probability (Appendix 2).

We used Python to implement the algorithm and Tensorflow (*Abadi, 2015*) for the deep-learning models, Scikit-learn (*Pedregosa, 2011*) for machine learning, and Scikit-image (*van der Walt et al., 2014*) for image processing. The code can be run either directly from Python or as an HTTP server, which enables access from other languages, such as Matlab. Scripts automate the training process, including optimising the hyperparameters, for the size categories and CNN architecture, and post-processing parameters (Appendices 1 and 2).

## Training data

Training data for the segmentation and bud assignment models comprises bright-field time-lapse images of yeast cells and manually curated annotations: a bit-mask outline for each cell (a binary image with the pixels constituting the cell marked with ones) and its associated tracking label and lineage assignment, if any. For the models optimised for microfluidic devices with traps, including both the single and five Z-section models, we took training images with five Z sections using a 60× lens. These images were from six independent experiments and annotated by three different people

and include a total of 3233 annotated cell outlines distributed across 1028 time points, 130 traps, and 28 fields-of-view. We include examples taken using cameras with different pixel sizes (0.182 μm and 0.263 μm). Cells in the training data were all derivatives of BY4741 growing in SC with glucose as carbon source. Most of the training images are of cells trapped in ALCATRAS devices (*Crane et al., 2014*), but some were for different trap designs. When training for a single Z-section, each of the five Z sections is independently presented to the CNN.

We split the training data into training, validation, and test sets (*Goodfellow et al., 2016*). We use the training set (588 trap images) to train the CNN and the validation set (248 trap images) to optimise hyperparameters and post-processing parameters. We use the test set (192 trap images) only to assess performance and generalisability after training. To increase the independence between each data set, our code allocates images using trap identities rather than time points or Z sections.

For the model optimised for microcolonies (*Figure 3—figure supplement 2*), we supplemented the ALCATRAS trap training set with 18 images from three fields-of-view (6% of the full data set) taken from the YeaZ bright-field training data (*Dietler et al., 2020*). To allow for overlaps in this data set, we re-annotated each field-of-view using our GUI (Appendix 4).

For training the tracking model, we used both the annotations from the segmentation training data, which are short time series of around five time points, and an additional data set of 300 time points of outlines, segmented using BABY and crudely tracked and then manually curated.

## Evaluating performance

### Segmentation

We evaluated BABY's segmentation on the training data's test set and compared with recent algorithms for processing yeast images (*Padovani et al., 2022*): Cellpose version 2.1.1 (*Stringer et al., 2021*; *Pachitariu and Stringer, 2022*), YeaZ (*Dietler et al., 2020*) from 11 October 2022, and our previous segmentation algorithm DISCO (*Bakker et al., 2018*). For Cellpose and YeaZ, we also trained new models on the images and annotations from both our training and validation sets, following their suggested methods (*Pachitariu and Stringer, 2022*; *Dietler et al., 2020*). Because neither handles overlapping regions, we applied a random order to the cell annotations such that pixels in regions of overlap were assigned the last observed label. We augmented the input data for each model by resampling the images five times, thus avoiding bias by forcing the models to adapt to uncertainty in the regions of overlap.

We assessed performance by calculating the intersection over union (IoU) of all predicted masks with the manually curated ground-truth masks from our test set. We paired predicted masks with the ground truth masks beginning with the highest IoU score; we assigned unmatched predictions an IoU of zero. To calculate the average precision for each annotated image, we used the area under the precision-recall curve for varying thresholds on the IoU score (*Manning et al., 2008*). Not all of the algorithms we tested give a confidence score, and so we generated precision-recall curves assuming ideal ordering of the predicted masks, by decreasing IoU. For the BABY models, ordering by mask probability produces similar results. We report the mean average precision over all images in the test set. To evaluate segmentation on microcolony images, we performed a similar analysis using the ground-truth annotations of the YeaZ bright-field training data (*Dietler et al., 2020*), but excluding the 18 images annotated and used to train BABY. We also re-trained the Cellpose and YeaZ models using our training data set supplemented with the microcolony images and evaluated the pre-trained bright-field YeaZ model, which includes this evaluation data in its training set, and the general-purpose pre-trained cyto2 Cellpose model, which segments cells from multiple different organisms.

### Tracking

We evaluated tracking on independent, manually curated data, comprising time series with 180–300 time points for 10 randomly selected traps from two experiments and four different growth conditions, making a total of 128 tracks. We initially generated the annotations using an early version of our segmentation and tracking models, but we manually corrected all tracking and lineage assignment errors and any obviously misshapen, misplaced or missing outlines, including removing false positives and adding outlines to the first visible appearance of buds. Unedited outlines, however, remain and will inevitably impart a bias. By requiring a mask IoU score of 0.5 or higher to match masks for the tracking, we expect to negate this bias. We compared BABY with YeaZ (*Dietler et al., 2020*) and

btrack (*Ulicna et al., 2021*) because Cellpose cannot track. For YeaZ, we used the model trained on our data; for btrack, we used the Cell-ACDC platform (*Padovani et al., 2022*) to combine segmentation by Cellpose with tracking by btrack.

The output of each model comprises masks with associated labels. We matched predicted and ground-truth masks at each time point to obtain maps from predicted to ground-truth labels, in descending order of mask IoUs but providing the mask IoU was greater than 0.5. We then calculated a track IoU between all predicted and ground-truth tracks: the number of time points where a predicted label mapped to a ground-truth label divided by the number of time points for which either track had a mask. This approach gave a map between predicted and ground-truth tracks in descending order of track IoUs. Using the mapping, we reported either the fraction of predicted tracks whose duration, the number of masks identified within that track, matched the ground-truth tracks (*Figure 3c*) or the distribution of track IoUs for all ground-truth tracks (*Figure 3—figure supplement 4*). For the Multiple Object Tracking Accuracy (MOTA) metric (*Bernardin et al., 2006*), we used the mask IoU to measure distance and considered correspondences as valid if the mask IoU ≥ 0.5.

## Assigning lineages

We evaluated BABY's lineage assignment using the lineage annotations included in the tracking evaluation data. These assignments pair bud and mother track labels. We used the track IoU to match ground-truth and predicted tracks above a given track IoU threshold and then compared lineage assignments based on this map. We counted true positives for each ground-truth bud-to-mother mapping if the ground-truth bud track had a matching predicted track and this predicted track had a predicted mother track matching the ground-truth mother track. False negatives were any ground-truth mother-bud pairs not counted as true positives; false positives were any predicted mother-bud pairs that were not counted as true positives. We repeated this analysis only for buds assigned to the central trapped cell or its matching predicted track.

## Estimating growth rates

We evaluated how well BABY estimates growth rates on independent, manually curated data comprising annotated time series of mother-bud pairs. We did not include this image data, which has growth in glucose, raffinose, pyruvate, and palatinose, in our training data. To select positions, traps, and time points, we randomly selected mother-bud pairs, rejecting samples only if there was no pair with a complete bud-to-bud cycle. We segmented this data with BABY and Cellpose and YeaZ trained on our data. To avoid penalising YeaZ and Cellpose for tracking errors, we found the matching predicted outlines with highest positive IoU for each ground-truth mask. We then used our method to estimate volumes (Appendix 3) to derive volumes for all masks, both ground-truth and predicted. Associating the masks with the ground-truth track, we fit a Gaussian process to each time series of volumes, omitting any time points with no matching mask. From the Gaussian process, we estimated a growth rate for each time point. Finally, we calculated the Root Mean Square Distance (RMSD) between the predicted and ground-truth estimates.

## Acknowledgements

We gratefully acknowledge support from the Leverhulme Trust (JMJP & PSS — grant number RPG-2018–04), the BBSRC (IF, IBNC, & PSS — grant number BB/R001359/1), and the European Union's Horizon 2020 research and innovation programme under the Marie Skłodowska Curie grant agreement no. 764591 — SynCrop (AFM & DYA).

---

## Additional information

### Funding

| Funder | Grant reference number | Author |
|---|---|---|
| Leverhulme Trust | RPG-2018-04 | Peter S Swain<br>Julian MJ Pietsch |

---

| Funder | Grant reference number | Author |
|---|---|---|
| Biotechnology and Biological Sciences Research Council | BB/R001359/1 | Peter S Swain<br>Ivan BN Clark<br>Iseabail Farquhar |
| Marie Sklodowska-Curie Actions | 764591 - SynCrop | Alán F Muñoz<br>Diane-Yayra A Adjavon |

The funders had no role in study design, data collection and interpretation, or the decision to submit the work for publication.

### Author contributions

Julian MJ Pietsch, Conceptualization, Software, Investigation, Visualization, Methodology, Writing - original draft, Writing – review and editing; Alán F Muñoz, Diane-Yayra A Adjavon, Software, Investigation, Visualization, Methodology, Writing – review and editing; Iseabail Farquhar, Methodology, Writing – review and editing; Ivan BN Clark, Investigation, Visualization, Methodology, Writing – review and editing; Peter S Swain, Conceptualization, Supervision, Funding acquisition, Writing - original draft, Writing – review and editing

### Author ORCIDs

Julian MJ Pietsch http://orcid.org/0000-0002-9992-2384
Peter S Swain http://orcid.org/0000-0001-7489-8587

### Decision letter and Author response

Decision letter https://doi.org/10.7554/eLife.79812.sa1
Author response https://doi.org/10.7554/eLife.79812.sa2

## Additional files

### Supplementary files

• MDAR checklist

### Data availability

Data is available at https://doi.org/10.7488/ds3427 and code from https://git.ecdf.ed.ac.uk/swain-lab/baby (copy archived at *Pietsch, 2023*).

The following dataset was generated:

| Author(s) | Year | Dataset title | Dataset URL | Database and Identifier |
|---|---|---|---|---|
| Pietsch JMJ | 2022 | A label-free method to track individuals and lineages of budding cells | https://doi.org/10.7488/ds3427 | Edinburgh DataShare, 10.7488/ds3427 |

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

## Appendix 1

### The BABY algorithm: identifying cells and buds

### Mapping cell instances to a semantic representation

For epifluorescence microscopy, samples are typically prepared to constrain cells in a monolayer. For cells with similar sizes that match the height of this constraint, they will be physically prevented from overlapping. If cells are of different sizes, however, then a small cell can potentially fit in gaps and overlap with others. This phenomenon is especially prevalent for cells that divide asymmetrically, where a small bud grows out of a larger mother.

Few segmentation algorithms identify instances of overlapping cells. Most, including recent methods for budding yeast (*Wood and Doncic, 2019*; *Dietler et al., 2020*; *Lugagne et al., 2018*), assume that cells can be labelled semantically, with each pixel of the image identified with at most one cell. Similarly, most tools for annotating also label semantically, and consequently curated training data does not allow for overlaps (*Dietler et al., 2020*), even when the segmentation algorithm could (*Lu et al., 2019*). Our laboratory's previous segmentation algorithm included limited overlap between neighbouring cells (*Bakker et al., 2018*), but not the substantial overlap we see between the smaller buds and their neighbours.

### Separating cells by size to disjoin overlapping cells

We rely on two consequences of the height constraint to segment overlapping instances. First, cells of different sizes show different patterns of overlap; second, the cells' centres are rarely coincident. Very occasionally, we do observe small buds stacked directly on top of each other, but neglecting these rare cases does not degrade performance. We therefore use morphological erosions to obtain semantic images by shrinking cell masks within a size category and, later, morphological dilations to approximate the original cell outlines from each resulting connected region.

To separate overlapping cells, we define three size categories and treat instances in each category differently. *Appendix 1—figure 1* illustrates our approach, where we segment a bud (orange outline) that overlaps a mother cell (green outline). The bud is only visible in the third and fourth Z sections of the bright-field images (*Appendix 1—figure 1a*). If used for training, we would split the manually curated outlines in this example (*Appendix 1—figure 1b*) into different size categories (*Appendix 1—figure 1c*). The bud is assigned to the small category. When we fill the outlines in this category and convert the image to a binary one (*Appendix 1—figure 1d*), the individual cell masks are distinct. For the large category, however, the masks are not separable when immediately converted, but become so when the filled outlines are morphologically eroded (*Appendix 1—figure 1d*). The largest size category tolerates more erosions than smaller ones, for which the mask may disappear or lose its shape.

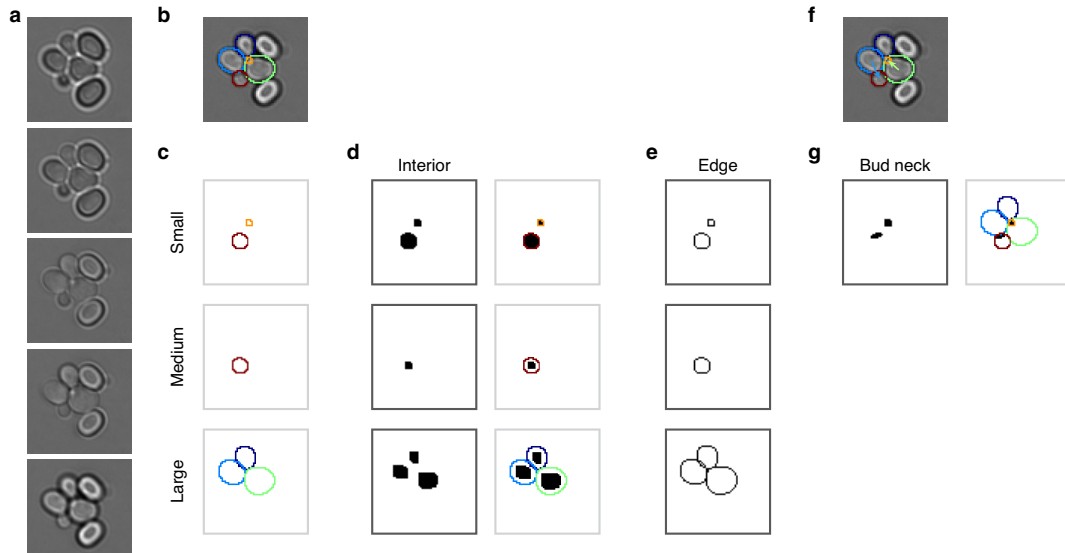

**Appendix 1—figure 1.** Mapping cell instances to semantic targets of a CNN. (**a**) Bright-field Z-sections of cells trapped in an ALCATRAS device. (**b**) Curated cell outlines overlaid on one bright-field section. (**c**) BABY separates outlines into categories by size, with each category having some overlap with neighbouring ones. Here the red outline in the medium category appears too in the small category. (**d**) Cell-interior targets for the CNN are the cell masks generated after different rounds of morphological erosions appropriate for each size category: no erosion for small cells, four iterations for medium, and five for large. On the right, we show the outlines overlaid on the target masks. (**e**) The CNN's edge targets are the outlines for each size category. (**f**) The curated cell outlines of b, but with arrows to show the lineages assigned during curation. (**g**) Using these curated lineages, we define the CNN's 'bud neck' target as the overlap of the bud mask with a morphological dilation of the mother mask (right).

## Determining the size categories

Using the training data – curated masks for each cell present at each trap at each time point, we identify the size categories that best separate overlapping cells. To begin, we calculate the overlap fraction – the intersection over union – between all pairs of cell masks. Its distribution reveals that the most substantial overlaps occur between cells of different sizes (*Appendix 1—figure 2a* – upper triangle).

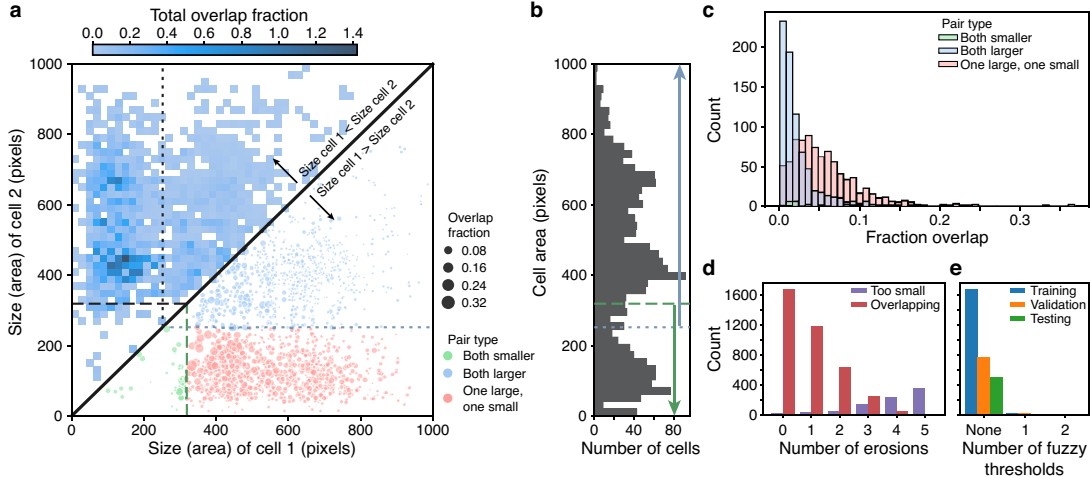

**Appendix 1—figure 2.** BABY reduces overlaps between cells through categorising cells by size. (**a**) Upper triangle: plotting the overlap fraction for each pair of cells – the intersection over union of their bit masks, shows that the majority of overlaps occur for cells of different sizes. Almost all overlaps have the size of cell 2 greater than the size of cell 1 and lie off the diagonal. Lower triangle: With a single fuzzy size threshold, cells in the small

*Appendix 1—figure 2 continued on next page*

*Appendix 1—figure 2 continued*

category have sizes less than the upper threshold ($T + P$; dashed line), and cells in the larger category have sizes greater than the lower threshold ($T - P$; dotted line). We show the overlap fraction by the size of the dot. Within each category (green and blue dots), small overlap fractions dominate; between the two categories (red dots), large overlaps dominate. By converting the bit masks into two binary images, one for each size category, rather than a single binary image, we therefore eliminate most of the substantial overlaps. (**b**) The distribution of all mask areas in the same training data for comparison. We indicate the size thresholds as in a. (**c**) The distributions of overlap fractions for mask pairs grouped using the fuzzy size threshold of a. We omit pairs that do not overlap for clarity. (**d**) Applying morphological erosions of the cell masks reduces the number of overlapping cell pairs, but generates smaller masks. We judge masks with areas below 10 pixels squared to be too small. (**e**) The numbers of overlapping cell pairs remaining from the training, validation, and test sets either before, denoted None, or after splitting into size categories and applying an optimised number of erosions.

We therefore choose the size categories so that most overlaps occur between pairs of cells in different categories and little overlap occurs between pairs of cells within a category. For example, rather than converting the cell masks directly into a single binary image for training, if first we divide cells into two size categories and convert the masks within each category to a separate binary image, giving two images rather than one, then in these two images we have eliminated all overlaps occurring between cells in the smaller category with cells in the larger category (*Appendix 1—figure 1* and *Appendix 1—figure 2* – lower triangle).

To divide the cell masks into two categories, we define a fuzzy size threshold using a threshold $T$ and padding value $P$. The set of smaller masks is all masks whose area is less than $T + P$; the set of larger masks is all masks whose area is greater than $T - P$. Consequently, the same mask can be in both sets (*Appendix 1—figure 1c*). This redundancy ensures the CNN produces confident predictions even for cells close to the size threshold, and we eliminate any resulting duplicate predictions in post-processing. BABY prevents a pair of masks overlapping by converted each into distinct binary images if the padded threshold separates their sizes: the smaller cell must have a size $< T - P$ and the larger cell must have a size $> T + P$. To scale with pixel size, we set $P$ to be 1% of the area of the largest mask in the training set.

To determine an optimal fuzzy threshold, we test $B = 100$ values evenly spaced between the minimal and maximal mask sizes and choose the threshold that minimises the summed overlap fraction for all mask pairs not excluded by the threshold. Even with one fuzzy threshold (*Appendix 1—figure 2a*), we exclude most of the pairs with substantial overlap – typically buds with neighbouring cells (*Appendix 1—figure 2c*).

After applying the threshold, overlaps between cells within a size category remain, and we reduce such overlaps using morphological erosions (*Appendix 1—figure 1*). We use the training data to optimise the number of erosions per size category. As the number of iterations increases, there is a trade-off between the number of overlapping mask pairs and the number of masks whose eroded areas become too small to be confidently predicted by the CNN (*Appendix 1—figure 2d*). Without erosion, the large cells can show overlaps; with too much erosion, the smallest masks distort their shapes or disappear. We therefore optimise the number of iterations separately for each size category, picking the highest number of iterations that do not let any of that category's training masks either fall below an absolute minimal size, defined as 10 pixels squared, or fall below 50% of the category's median cell size before any erosions.

Combining categorising by size with eroding reduces the number of pairs of overlapping masks almost to zero (*Appendix 1—figure 2e*). We arrive at three size categories by first introducing an additional fuzzy threshold for each of the two initial size categories. These thresholds are similarly determined by testing $B = 100$ fuzzy threshold values and calculating the overlap fraction for all mask pairs not excluded by either the original or the new threshold. We only keep one of the new thresholds – the one minimising the overlap fraction, giving three size categories in total. This extra category results in a further, although proportionally smaller, decrease in the number of overlapping masks.

After erosion, mask interiors within each size category are easily identified, but with less resolved edges. To help alleviate this loss, we generate edge targets for the CNN from the training data (*Appendix 1—figure 1e*) – the outlines of all cells within each size category.

The microcolony training images for YeaZ (*Dietler et al., 2020*) include a larger range of cell sizes than in our training set. We therefore increased $B$ to 200 (*Figure 3—figure supplement 2*) and

determined the thresholds on square-root transformed sizes. We transformed these thresholds back to the original scale when providing targets for the CNN.

## Four types of training targets

We further annotate the curated data with lineage assignments (*Appendix 1—figure 1f*), which BABY uses to generate 'bud neck' targets for the CNN (*Appendix 1—figure 1g*). The final target is another binary image, which is only true wherever masks of any size overlap.

In total, the eight training targets for the CNN are the mask interiors and edges for three size categories, the bud necks, and the overlap target. We weighted the targets according to their difficulty and importance in post-processing steps: the large and medium edge targets and small interior target with a weight of two and the small edge target with one of four.

## Predicting semantic targets with a convolutional neural network

We trained fully convolutional neural networks (*Goodfellow et al., 2016*) to map a stack of bright-field sections to multiple binary target images. We show some example inputs and outputs in *Appendix 1—figure 1*, but we also trained networks with only one or three bright-field sections. The intensities of the bright-field sections were normalised to the interval $[-1, 1]$ by subtracting the median and scaling according to the range of intensities expected between the 2nd and 98 percentiles.

Each output layer of the CNN approximates the probability that a given pixel belongs to the target class, being a convolution with kernel of size $1 \times 1$ and sigmoidal activation. All other convolutions had kernels of size $3 \times 3$ with ReLU activation and used padding to ensure consistent dimensions for input and output layers.

## Augmenting the training data

To prevent over-fitting and improve generalisation, we augmented the training data (*Goodfellow et al., 2016*). Each time the CNN sees a training example, it sees too a randomly selected series of image manipulations applied to the input and target. The same training example therefore typically appears differently for each epoch.

Three augmentations were always applied and the others applied with a certain probability. The fixed augmentations were horizontal and vertical translations and if the bright-field input had more Z sections than expected by the network, we selected a random subset, excluding any subsets with selected sections separated by two or more missing sections. Those augmentations applied with a probability $p$ comprised elastic deformation ($p = 0.35$), image rotation ($p = 0.35$), re-scaling ($p = 0.35$), vertical and horizontal flips (each with $p = 0.35$), addition of white noise ($p = 0.35$), and a step shift of the Z sections ($p = 0.35$). The probability of not augmenting was thus $p = 0.05$. To show a different region of each image-mask pair at each epoch, translation, rotation, and re-scaling were all applied to images and masks before cropping to a consistent size ($128 \times 128$ pixels for a pixel size of 0.182μm). Using reflection to handle the boundary, translations were over a random distance and rotations over a random angle. To apply elastic deformation, as described for the original U-Net (*Ronneberger et al., 2015b*), we used the elasticdeform package (*van Tulder, 2022*) for an evenly spaced grid with target distance between points of 32 pixels and standard deviation of displacement of 2. Augmentation by re-scaling was for a randomly selected scaling factor up to 5%. Augmentation by addition of white noise involved adding random Gaussian noise with a standard deviation picked from an exponential distribution with rate  to each pixel of the (normalised) bright-field images.

To reduce aliasing errors when manipulating binary masks during augmentation, we applied all image transformations independently to each filled mask before converting the transformed masks into one binary image. Further, before a transformation, we smoothed each binary filled outline with a 2D Gaussian filter and found the transformed binary outline with the Canny algorithm. To determine the standard deviation of this Gaussian filter, σ, we tested a range of values on the training outlines. For each filled outline and σ, we applied the filter followed by edge detection and filling. We then calculated the intersection over union of the resulting filled outline with the original filled outline. We observed that as a function of edge length, defined as the number of edge pixels, the σ producing the highest intersection over union increased exponentially. We consequently used an exponential fit of this data to estimate an appropriate σ for each outline.

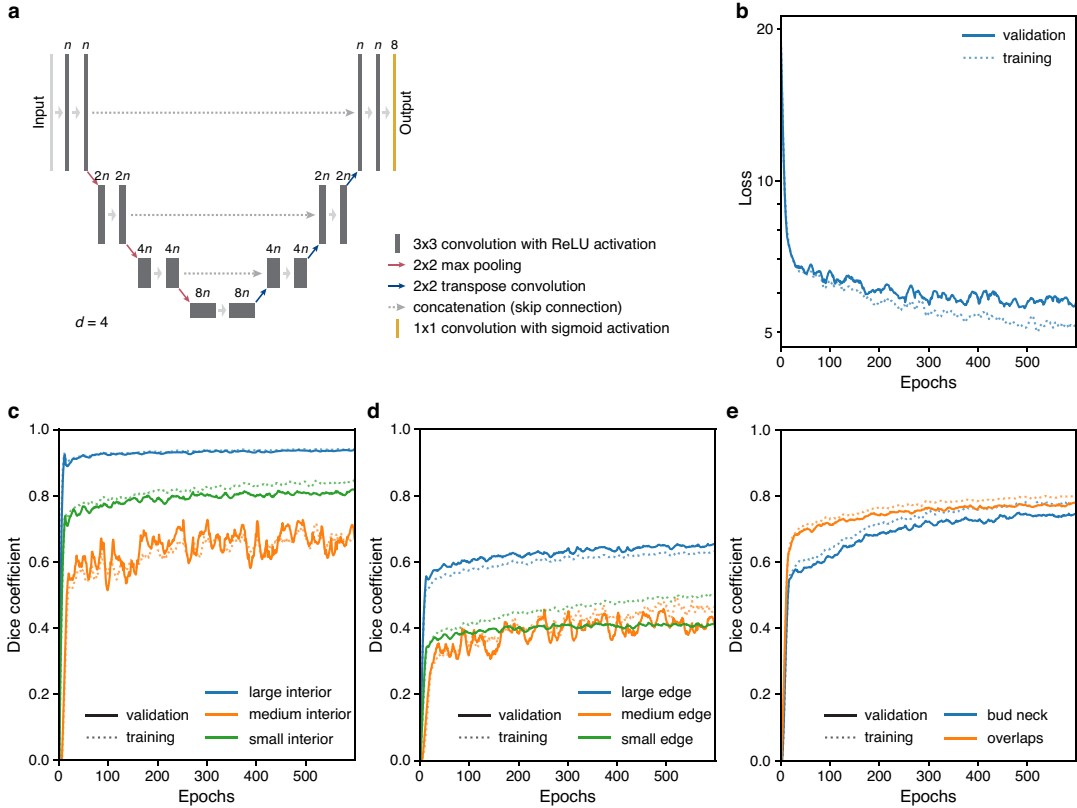

**Appendix 1—figure 3.** Training performance of the multi-target U-Net. (**a**) A schematic of a U-Net architecture with depth $d = 4$. The labels above the convolution operations indicate the number of output filters as a multiple of $n$. Layer heights indicate reduction in image size with network depth. (**b**) Loss for the fully trained 5Z model U-Net with hyperparameters chosen from training trial giving the lowest final validation loss: a U-Net with depth $d = 4$, filter factor $n = 16$, and batch normalisation. (**c–e**) Performance of (**c**) interior, (**d**) edge and (**e**) bud neck, and overlap targets by the U-Net of b decomposed into the three different size categories when possible. The Dice coefficient reports similarity between prediction probabilities and target masks with a value of 1 indicating identity. For two sets $X$ and $Y$, the Dice coefficient is $\frac{2|X \cap Y|}{|X|+|Y|}$.

## Training

We trained networfks using Keras with TensorFlow 2.8. We used Adam optimisation with the default parameters except for a learning rate of 0.001 and regularised by keeping only the network weights from the epoch with the lowest validation loss (similar in principle to the early stopping method) (*Goodfellow et al., 2016*). We train for 600 epochs, or complete iterations over the training data set.

The loss function is the sum of the binary cross-entropy and one minus the Dice coefficient across all targets:

$$L = -\sum_{i}\left[y_i \log \hat{y}_i + (1 - y_i)\log(1 - \hat{y}_i)\right] + 1 - \frac{2\sum_i y_i \hat{y}_i}{\sum_i y_i + \sum_i \hat{y}_i} \tag{1}$$

where $y$ is the tensor of true values, $\hat{y}$ is the CNN's sigmoid tensor output of the CNN, and $i$ is a vectorised index.

Each CNN is trained to a specific pixel size, and we ensured that training images and masks with different pixel sizes were re-scaled appropriately

## CNN architectures

We trialled two core architectures for the CNN – U-Net (*Ronneberger et al., 2015b*; *Appendix 1—figure 3a*) and Mixed-Scale-Dense (MSD) (*Pelt and Sethian, 2018*) – and optimised hyperparameters to find the smallest loss on the validation data.

The U-Net performed best (see 'Optimising hyperparameters' below). The U-Net has two parts: an encoder that reduces the input into multiple small features and a decoder that scales these features up into an output (**Ronneberger et al., 2015b**). Each step of the encoder comprises a convolutional block, which creates a new, larger set of features from its input. To force the network to keep only small, relevant features, a down-sampling step is applied after three convolutional blocks. This maximum pooling layer reduces the size of the features by half by replacing each two-by-two block of pixels by their maximal value. The decoder also comprises convolutional blocks, but with up-sampling instead of down-sampling. The up-sampling step is the inverse of down-sampling: each pixel is turned into a two-by-two block by repeating its value. Finally, most characteristic of the U-Net is its skip layers. These layers preserve information on the global organisation of the pixels by passing larger-scale information from the encoder to the decoder after each up-sampling step. They act by concatenating the same-size layer of the encoder into the decoder layers, which are then used as inputs for the next step of the decoder. The decoder is therefore able to create an output from both the local features that it up-sampled and from the global features that it obtains from the skip layers.

For the U-Net, we optimised for depth, for a scaling factor for the number of filters output by each convolution, whether or not to include batch normalisation, and for the proportion of neurons to drop out on each batch. For the MSD, we optimised for depth, defined as the total number of convolutions, for the number of dilation rates to loop over with each loop increasing dilation by a factor of two, for an overall dilation-rate scaling factor, and whether or not to include batch normalisation.

## Optimising hyperparameters

We used KerasTuner with TensorFlow 2.4 to optimise hyperparameters, choosing random search with default settings, training for a maximum of 100 epochs, and having 10 training and validation steps per epoch. The U-Net and MSD networks with the lowest final validation loss were then re-trained as described, and the network with the lowest validation loss chosen.

For our data, the best performing model was a U-Net with depth four, and so three contractions, with a scaling factor of 16 for the number of filters output by each convolution, giving 16, 32, 64 and 128 filters for each of the two chained convolution layers of the encoding and decoding blocks, with batch normalisation, and with no drop-out. We show its performance for the 5Z model in **Appendix 1—figure 3c–e**.

## Identifying cells

To identify cell instances from the semantic predictions of the CNN, we developed a post-processing pipeline with two parts (**Appendix 1—figure 4a**): proposing unique cell outlines and then refining their edges.

The pipeline includes multiple parameters that we optimise on validation data by a partial grid search. We favour precision, the fraction of true predicted positives, over recall, the fraction of ground truth positives we predict, by maximising the $F_\beta$ score with $\beta = 0.5$. Recall that for true positives TP, false negatives FN, and false positives FP,

$$F_\beta = \frac{(1 + \beta^2)\text{TP}}{(1 + \beta^2)\text{TP} + \beta^2\text{FN} + \text{FP}}. \tag{2}$$

We measure how well two masks match using the intersection over union (IoU) and consider a match to occur if $\text{IoU} > 0.5$. Nevertheless, multiple predictions may match a single target mask because predicted masks can overlap too. We therefore count true positives as target masks for which there is at least one predicted mask with $\text{IoU} > 0.5$. Any predicted masks in excess of the true positive count are false positives, thus avoiding double counting. Unmatched target masks are false negatives.

## Proposing cell outlines

The post-processing pipeline starts by identifying candidate outlines independently for each size category. The CNN's outputs are images $p_{xy}^{(S,C)} \in [0, 1]$ approximating the probability that a pixel at position $(x, y)$ belongs to either the small, medium, or large size categories, denoted $S$, and to one

of the other classes, denoted $C$: either the interior (*Appendix 1—figure 4b*), edge (*Appendix 1—figure 4e and f*), bud neck, or general overlap classes.

In principle, we could find instances for each size category by thresholding the interior probability $p_{xy}^{(S,\text{interior})}$ and identifying connected regions as outlines. To further enhance separability, however, we also re-weight the interior probabilities using the edge probabilities. Specifically, we identify connected regions from semantic bit masks $b_{xy}^{(S,\text{interior})}$ by those pixels that satisfy

$$p_{xy}^{(S,\text{interior})} \left[ 1 - \text{Dilate}^{N_{\text{dilate}}} \left( p_{xy}^{(S,\text{edge})} \right) \right] > T_{\text{interior}} \tag{3}$$

where $\text{Dilate}^N$ specifies $N$ iterations of a gray scale morphological dilation and $T_{\text{interior}}$ is a threshold. We optimise the thresholds $T_{\text{interior}} \in [0.3, 0.95]$, number of dilations $N_{\text{dilate}} \in \{0, 1, 2\}$, and the order of connectivity (one- or two-connectivity) for each size category.

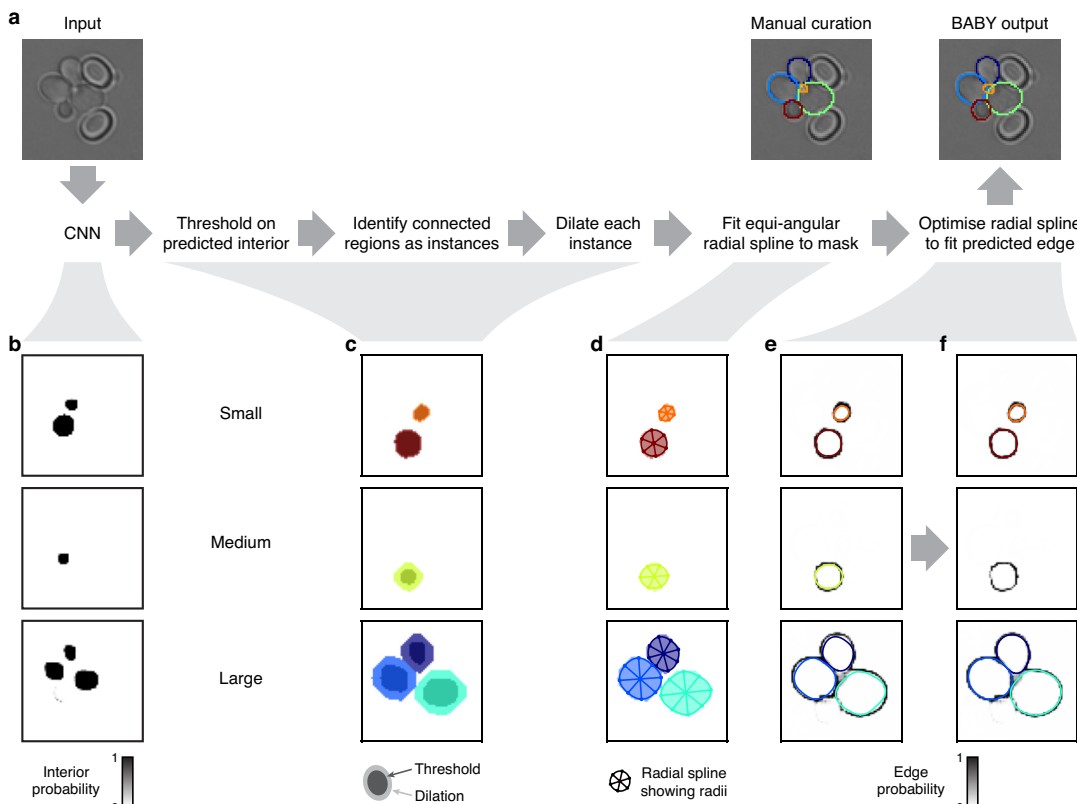

**Appendix 1—figure 4.** Segmenting overlapping cell instances from the CNN's output. (**a**) A flow chart summarising the post-processing for identifying individual instances from the CNN's multi-target output. Here and below, we show results using the five Z sections of *Appendix 1—figure 1* as input to the CNN, and one of which we repeat here. (**b**) The probability maps output by the CNN for the interiors of small, medium, and large cells. (**c**) Bit masks obtained by thresholding on the CNN's output. Darker shading shows bit masks before we dilate each instance to compensate for the erosion applied when generating the training targets. Colour indicates distinctly identified instances. (**d**) We show the initial, equiangular radial splines proposed for each instance overlaid on the dilated bitmasks from c, with the rays defining placement of the knots as spokes. (**e**) The same initial proposed radial splines overlaid on the edge target probability maps output by the CNN. (**f**) The radial splines after optimisation to match edge probabilities. The outline in the medium size category is detected as a duplicate and not optimised.

The connected regions in $b_{xy}^{(S,\text{interior})}$ define masks that are initial estimates of the cells' interiors (darker shading in *Appendix 1—figure 4c*). We generate the cell interiors for training the CNN by iterative, binary morphological erosions of the full mask, where the number of iterations $N_{\text{erosion}}$ is pre-determined for each size category. First, we remove small holes and small foreground features by applying up to two binary morphological closings followed by up to two binary morphological

openings. Second, we estimate full masks $b_{xy}^{(S,\text{full})}$ from each putative mask by applying $N_{\text{erosion}}$ binary dilations (light shading in *Appendix 1—figure 4c*) undoing the level of erosion on which the CNN was trained. We optimise both the numbers of closing, $N_{\text{closing}}$, and opening, $N_{\text{opening}}$, operations.

Any masks whose area falls outside the limits for a size category, we discard. For each category, however, we soften the limits, on top of the fuzzy thresholds, by optimising an expansion factor $F_{\text{exp}} \in [0, 0.4]$, which extends the limits by a fractional amount of that category's size range. We also optimise a single hard lower threshold $T_{\text{min}} \in [0, 20]$ on mask area.

## Using splines to describe mask edges

To prepare for refining edges and to further smooth and constrain outlines, we use a radial spline to match the edge of each of the remaining masks (*Appendix 1—figure 4d*). As in DISCO (*Bakker et al., 2018*), we define radial splines as periodic cubic B splines using polar coordinates whose origin is at the mask's centroid. We generalise this representation to have a variable number $n^{(S,i)}$ of knots per mask specified by $n^{(S,i)}$-dimensional vectors of radii $r^{(S,i)}$ and angles $\theta^{(S,i)}$:

$$\theta \mapsto s\left(\theta, r^{(S,i)}, \theta^{(S,i)}\right)$$
(4)

A mask's outline is then fully specified by those pixels that intersect with this spline.

To initially place the knots, we search along rays originating at the centroid of each mask $b_{xy}^{(S,\text{full})}$ and find where these rays intersect with the mask edge. We determine the edge by applying a minimum filter with two-connectivity to the mask and set to true all pixels in the filtered image that are different from the original one. We then smooth the resulting edge image using a Gaussian filter with $\sigma = 0.5$. For a given polar angle $\theta$, we find the radius of the corresponding knot by averaging the edge pixels that intersect with the ray, weighted by their values. We use the major axis of the ellipse with the same normalised second central moment as the mask (regionprops function from Scikit-image *van der Walt et al., 2014*) to determine both the number of rays, and so knots, and their orientations. The length $\ell^{(S,i)}$ of the major axis gives the number of rays: four for $0 < \ell^{(S,i)} < 5$; six for $5 \leq \ell^{(S,i)} < 20$; and eight for $\ell^{(S,i)} \geq 20$. For this initial placement, we choose equiangular $\theta^{(S,i)}$, with the first knot on the ellipse's major axis.

## Discarding poor or duplicated outlines

The quality of the outline masks $\hat{o}_{xy}^{(S,i)}$ derived from these initial radial splines are then assessed against the edge probabilities generated by the CNN (*Appendix 1—figure 4e*) and masks of poor quality discarded. We calculate the edge score for a given outline as

$$\eta^{(S,\text{edge},i)} = \frac{1}{N_x N_y} \sum_{xy} p_{xy}^{(S,\text{edge})} \text{Dilate}^2\left(\hat{o}_{xy}^{(S,i)}\right).$$
(5)

We discard those outlines for which the edge score is less than a threshold, where the thresholds $T_{\text{edge}} \in [0, 1)$ are optimised for each size category based on the range of edge scores observed.

With a smoothed and filtered set of outlines, we proceed by detecting and eliminating any outlines duplicated between size categories. We start by filling the outlines to form a set of full masks $\hat{m}_{xy}^{(S,i)}$. We then compare these masks between neighbouring size categories $S_j$ and $S_k$. We consider the pair of masks $i_1$ and $i_2$ as duplicates if one of the masks is almost wholly contained within the other:

$$\frac{\sum_{xy} \hat{m}_{xy}^{(S_j,i_1)} \cap \hat{m}_{xy}^{(S_k,i_2)}}{\min\left(\sum_{xy} \hat{m}_{xy}^{(S_j,i_1)}, \sum_{xy} \hat{m}_{xy}^{(S_k,i_2)}\right)} > T_{\text{containment}}$$
(6)

for some threshold $T_{\text{containment}} \in [0, 1]$, optimised on validation data. For pairs that exceed this threshold, we keep only the mask with the highest edge score given by *Equation 5*.

For each size category, the first part of the post-processing pipeline finishes with the set of outlines that pass these size, edge probability, and containment thresholds. *Appendix 1—table 1* gives values for the optimised post-processing parameters.

**Appendix 1—table 1.** Optimised post-processing parameters for BABY's standard model. The standard model takes five bright-field Z sections with a pixel size of 0.182m as input. Excepting

$T_{min}$ and $T_{containment}$, we optimised parameters separately for each size category.

| Parameter | $S$ = small | $S$ = medium | $S$ = large |
|---|---|---|---|
| $T_{interior}$ | 0.35 | 0.5 | 0.95 |
| $N_{dilate}$ | 0 | 0 | 0 |
| Connectivity | 2 | 1 | 1 |
| $N_{closing}$ | 0 | 0 | 0 |
| $N_{opening}$ | 0 | 0 | 2 |
| $F_{exp}$ | 0.32 | 0.06 | 0.28 |
| $T_{edge}$ | 0.0012 | 0.0028 | 0.0 |
| $T_{min}$ | | 19 | |
| $T_{containment}$ | | 0.85 | |

## Refining edges

The outlines $\hat{o}_{xy}^{(S,i)}$, defined by the radial splines, do not directly make use of the CNN's edge targets for their shape and deviate from $p_{xy}^{(S,\text{edge})}$, particularly for those in the large size category (*Appendix 1—figure 4e*).

We therefore optimise the radial splines to better match the predicted edge. This optimisation is challenging because $p_{xy}^{(S,\text{edge})}$ provide only a semantic representation of the edge – the association of a given pixel $(x, y)$ with a particular instance $i$ is unknown. Our approach is to use the outlines to generate priors on whether predicted edge pixels associate with a given instance. We then apply standard techniques to optimise the fit of the radii and angles of the knots for each outline's spline to its instance's likely edge pixels.

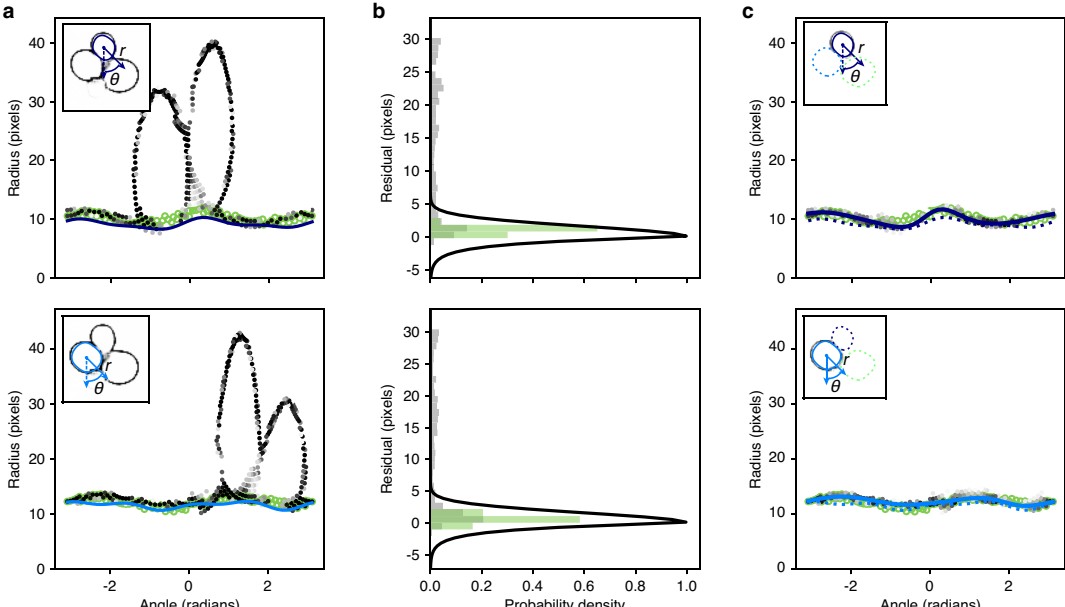

**Appendix 1—figure 5.** Optimisation of the radial spline to fit the predicted edge. (**a**) We show the rdge pixels predicted by the CNN in polar coordinates for two different instances in the top and bottom panels. Darker shading indicates a higher probability of being an edge. Open green circles are the manually curated ground truth. Solid lines are the initial radial splines estimated from the interiors predicted by the CNN. Insets show the predicted edge in cartesian coordinates with the instance providing the origin marked by its initial outline and the indicated polar coordinates. (**b**) We plot the binned residuals of the predicted edge pixels with the initial radial spline for the examples ofa. The algorithm considers only edge pixels with probability greater than 0.2. Binned

*Appendix 1—figure 5 continued on next page*

*Appendix 1—figure 5 continued*

residuals for the ground truth are in green. Black lines show the function used to re-weight pixel probabilities for each instance. (**c**) As for a, but after the edge pixels have been re-weighted for each instance. Solid lines indicate the optimised radial spline. We show the outline favoured by the instance-association probability as a solid line in the inset; disfavoured outlines are dashed.

To associate pixels with instances, we first calculate the radial distance of each pixel from the initial radial spline function $s$ proposed for an instance in *Equation 4*. To increase speed, we consider only pixels where $p_{xy}^{(S,\text{edge})} > 0.2$. Expressing the edge pixels in polar coordinates as $(\rho, \phi)$ with the origin at the instance's centroid, this distance is

$$R_{xy}^{(S,i)} = \rho - s\left(\phi, \boldsymbol{r}^{(S,i)}, \boldsymbol{\theta}^{(S,i)}\right) \tag{7}$$

which we will refer to as a pixel's residual. We give two examples of edge pixels (*Appendix 1—figure 5a*) and of the corresponding residuals (*Appendix 1—figure 5b*), which highlight the need to associate pixels with a given instance before attempting to optimise the spline.

We use the residuals, *Equation 7*, to assign prior weights to pixels:

$$\text{W(R)} = \begin{cases} e^{-R^2/\sigma_G} & \text{if } R \geq 0 \\ e^{R/\sigma_E} & \text{if } R < 0 \end{cases} \tag{8}$$

where $\sigma_G = 5$ and $\sigma_E = 1$. The function $W$ is a Gaussian function of the residual for pixels exterior to the proposed outline and an exponential function for pixels interior (*Appendix 1—figure 5b*). This asymmetry should increase tolerance for interior edge pixels, which may belong to neighbouring instances overlapping with the cell of interest. In such cases, we should thus improve instance association, particularly where the edges of each of the cells intersect.

With these prior weights, we find the probability that each edge pixel associates with a particular instance and not with the others via:

$$p_{xy}^{(S,\text{edge},i)} = p_{xy}^{(S,\text{edge})} \times W(R_{xy}^{(S,i)}) \times \left(1 - \frac{1}{n-1}\sum_{j\neq i} W(R_{xy}^{(S,j)})\right) \tag{9}$$

where $n$ is the number of detected instances in this and adjacent size categories, with $j$ running over all these instances. We filter the result, keeping only those edge pixels with $p^{(S,\text{edge},i)} > 0.1$. Examples are shown in *Appendix 1—figure 5c*.

We optimise the knot radii $\boldsymbol{r}^{(S,i)}$ and angles $\boldsymbol{\theta}^{(S,i)}$ for each radial spline by minimising the squared radial residual between the spline and the edge pixels, *Equation 7*. With residuals weighted by $p_{xy}^{(S,\text{edge},i)}$, *Equation 9*, and initial values taken from each $\hat{o}_{xy}^{(S,i)}$, we constrain radii to a 30% change from their initial values and angles to a change of ±25% of the initial angular separation between knots: $\theta_{i+1} - \theta_i$. The resulting optimised radial splines provide the outlines output by the BABY algorithm.

## Appendix 2

### The BABY algorithm: tracking cells and identifying lineages

To track cells and lineages, we have two tasks: first, link cell outlines from one time point to the next (*Appendix 2—figure 1a*), and second, identify mother-bud relationships (*Appendix 2—figure 1b*).

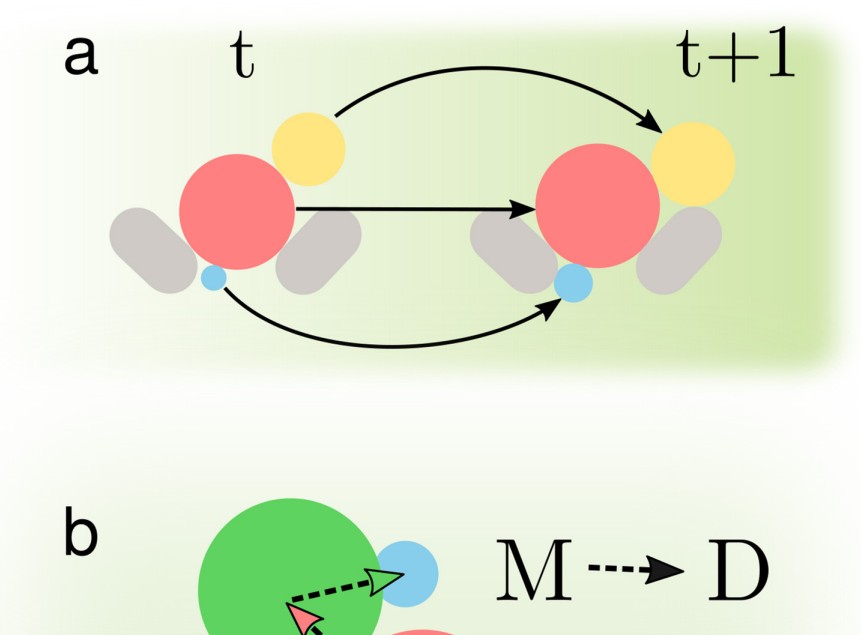

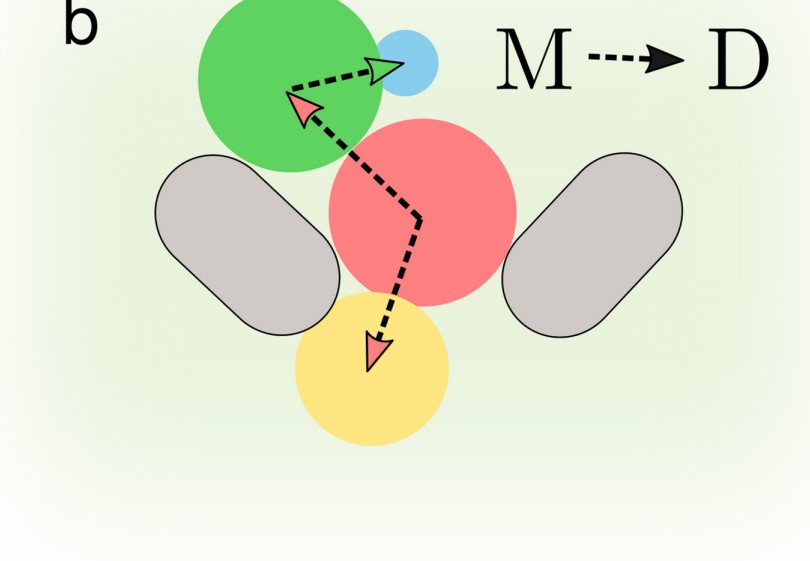

**Appendix 2—figure 1.** Determining accurate lineages requires solving two independent tasks. (**a**) We must identify cells across time points regardless of how they grow and move within the images. (**b**) We have to find the mother-bud relationship between cells at every time point.

## Tracking cells from image to image

In our setup, daughter cells may be washed out of the microfluidic device and so disappear from one time point to the next. These absences undermine other approaches to tracking, such as the Hungarian algorithm (*Versari et al., 2017*).

To track cells (*Appendix 2—figure 2*), we use the changes in their masks over time to indicate identity. From each mask, we extract an array of attributes, such as the mask's area, major axis length, etc., and to compare a mask at one time point to a mask at another time point, we subtract

the two corresponding arrays of features. This array of differences is the array of features we use for classification.

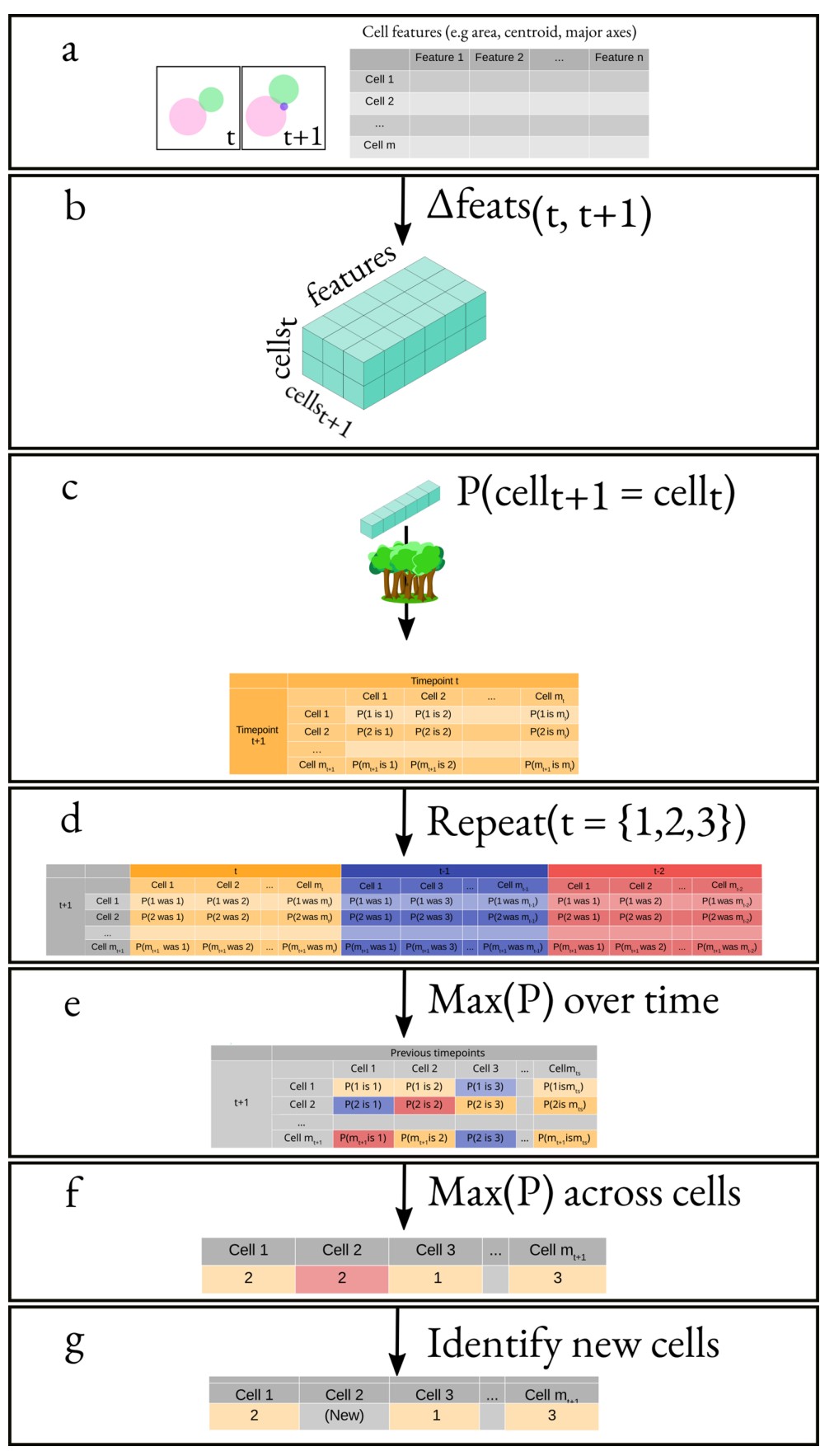

*Appendix 2—figure 2 continued on next page*

*Appendix 2—figure 2 continued*

**Appendix 2—figure 2.** Overview of the algorithm for tracking cells. (**a**) We obtain the attributes of all cells at times $t$ and $t + 1$. This results in two matrices of shape $(m_t, n)$ and $(m_{t+1}, n)$, where $m_x$ is the number of cells at time $x$ and $n$ is the number of attributes. (**b**) We generate a feature vector for every cell pair by subtracting, element-wise, the attributes of all cells at time $t$ from the attributes of all cells at time $t + 1$. (**c**) We apply a classifier to the feature vector corresponding to each pair of cells. (**d**) We repeat the same process but using $t - 1$ and $t - 2$ instead of $t$. (**e**) We pick the maximal probability for every pair of cells over all the probability matrices. (**f**) We apply our cell-labelling algorithm to assign cell pairs (**g**) Finally, we use a threshold to identify new cells.

Our training data comprises a series of manually labelled time-lapse images from four experiments. For two consecutive time points, we calculated the difference in feature arrays between all pairs of cells and grouped these difference arrays into two classes: one for identical cells – cells with the same label – and one for all other cells.

## Using multiple time points in the past:

To generate additional training data, we use multiple time points backwards in time. For example, for time $t$, we generate not only feature vectors by comparing with cells at $t - 1$, but also with cells at $t - 2$ and $t - 3$. We found this additional data increased generalisability, maintaining accuracy across a wider range of imaging intervals and growth rates. For the purpose of training, we treat the additional data as consecutive time points: the algorithm does not know whether the features come from one or more than one time point in the past.

As part of testing if all features contribute to the learning, we divided the features into two overlapping sets. One set had no features that explicitly depend on distance, comprising area, lengths of the minor and major axes, convex area, and area of the bounding box; the second set did include distance-dependent features, comprising area, lengths of the minor and major axes, and convex area again, but additionally including the mask's centroid, and the distance obtained from the x- and y-axis locations.

We compared three standard algorithms for classification (***Bishop, 2006***): the Support Vector Classifier (SVC), Random Forest, and Gradient Boosting, specifically Xtreme Gradient Boosting (***Chen and Guestrin, 2016***). We used scikit-learn (***Pedregosa, 2011***) to optimise over a grid of hyperparameters.

For the SVC, we considered a regularisation parameter $C$ of 0,1, 10, or 100; a $\Gamma$ kernel coefficient of 1, 10–3, or 10–4; no shrinking heuristic to speed up training; and either a radial basis function or sigmoid kernel.

For the Random Forest, we explored a range between 10 and 80 estimators and a depth between 2 and 10 levels.

For Gradient Boosting, we used a maximal depth of either 2, 4, or 8 levels; a minimal child weight of 1, 5, or 10; gamma, the minimal reduction in loss to partition a leaf node, of 0.5, 1, 1.5, 2, or 5; and a sub-sampling ratio of 0.6, 0.8, or 1.

Within the training data, the number of time points for each experiment is different. To prevent biases toward long experiments, we define the accuracy as the fraction of true positives – cells correctly linked between images – and compare the precision and recall of this time-averaged accuracy.

After training, we evaluated which features were important using the Random Forest. The distribution of the feature weights depends on whether we include distance (***Appendix 2—figure 3***), and excluding distance distributes the weights more evenly.

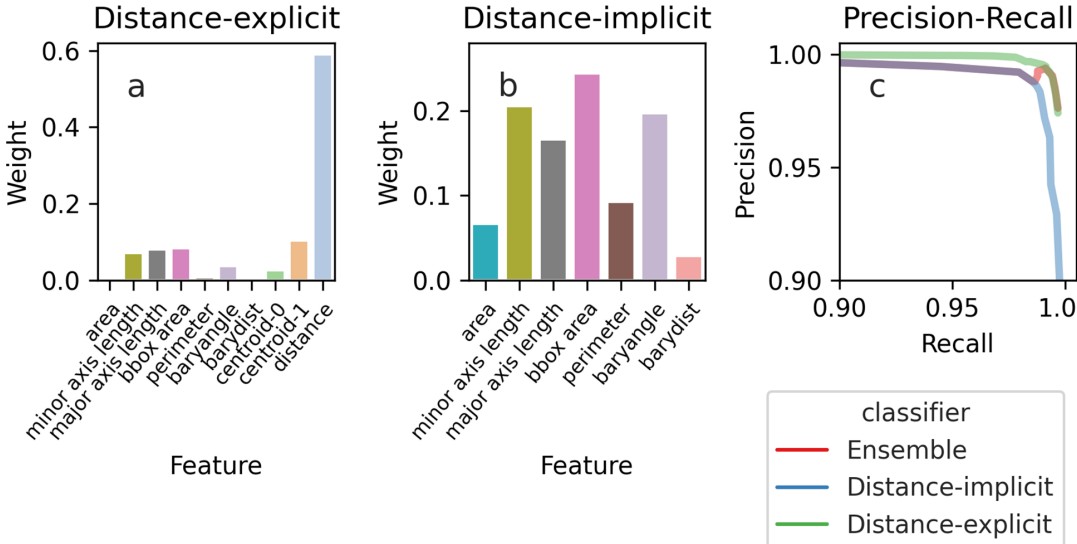

**Appendix 2—figure 3.** The importance of the features used by the Random Forest classifier for tracking cells between time points. Depending on the features we use, the feature weights, a measure of their importance, are more evenly spread. (**a**) If we train the classifier using features that explicitly include distance-dependence, distance drives the decisions, and the remaining features are only used for marginal cases. (**b**) If we train the classifier using distance-implicit features, however, the weights are more uniform. (**c**) The precision-recall curve shows high accuracy for both sets of features.

## An ensemble model

The precision-recall curve indicates that using the distance-explicit features is best, although both sets of features have high accuracy (*Appendix 2—figure 3c*). Despite performing better on our test data, we expect that using the distance-explicit features may perform worse if the cells pivot or become displaced. Therefore, we use the non-explicit features as our main model, but also use the distance-explicit features to resolve any ambiguous predictions. The ensemble model performs similarly to the distance-implicit classifier, but for more stringent thresholds behaves like the distance-explicit one.

## Making predictions

To predict with the classifier, we use data from the current time point and the two most recent previous time points. We generate feature arrays between $t$ and independently $t-1$ and $t-2$ and feed both arrays to the classifier. If the probability returned is greater than 0.9, we accept the result; if the probability lies between 0.1 and 0.9, we use instead the probability returned by the backup classifier, which uses the distance-explicit features.

Using multiple time points to track cells has two advantages: first, it reduces noise generated by artefacts, either in image acquisition, such as a loss of focus, or in segmentation; second, it ensures that cells are more consistently identified if their position or shape transiently changes. Including data further back in time is neither computationally efficient nor more accurate, and greater than three time points is long, over 15 minutes in our experiments and about a sixth of a cell cycle.

We apply the linear sum assignment algorithm, via SciPy, on the probability matrix of predictions to assign labels (Appendix 2 Algorithm 1). This approach guarantees at most one outline is assigned to each cell by choosing the set of probabilities whose total sum is highest. To match a cell with its previous self, we pick the cell in the recent past that generates the highest probability when paired with the cell of interest, providing this probability is greater than 0.5. We label a cell as new if the probabilities returned from pairing with all cells in the recent past is below 0.5.

Algorithm 1 **Cell labelling**

---

**Data:** *probMat, threshold, oldLabels, maxLabel*
**Result:** New cell labels (*newLabels*)
let *newLabels* be zeros(ncols(*probMat*));
**for** *old, new* ∈ *linearSumAssignment*(−*ProbMat*) **do**
 **if** *probMat*[*old, new*] > *threshold* **then**
 *newLabels*[*new*] ← *oldLabels*[*old*];
 **end**
**end**
**for** *label* ∈ *newLabels* **do**
 **if** *label*! = 0 **then**
 *label* = *maxLabel* + 1;
 *maxLabel* = label;
 **end**
**end**
return *newLabels*

---

## Assigning lineages

We wish to identify which cells are buds of mothers and which mothers have buds. This problem is analogous to tracking, but, rather than identifying pairs of cells that are the same cell at different time points, we must identify pairs of cells that are a mother-bud pair at one time point. We therefore seek to determine the probability that a pair of cells is a mother-bud pair (*Appendix 2—figure 4*). Unlike tracking, however, we anticipated that the cell outlines alone would be at best a weak indicator of this probability.

## Defining mother-bud features

We observed that cytokinesis is sometimes visible in bright-field images as a darkening of the bud neck and designed features to exploit this characteristic of mother-bud pairs.

Such features often rely on the CNN's prediction of bud necks. For generalisability and to avoid ambiguity, we chose to define the corresponding training target using manually annotated outlines and lineage relationships, rather than relying on a fluorescent bud-neck marker. Specifically, we define a binary semantic 'bud-neck' training target that is true only at pixels where a mother mask, dilated twice by morphological dilation, intersects with its assigned bud (*Appendix 1—figure 1*). Assigning a time of cytokinesis by eye is challenging, and so we included two constraints to identify a bud. First, the bud must be current – as soon as BABY finds another bud associated to the mother, we drop the current one. Second, we exclude buds if their area is larger than and has always been larger than 10 μm$^2$ (300 pixels for our standard training target with a pixel size of 0.182 μm and corresponding to a sphere of ~ 24 μm$^3$).

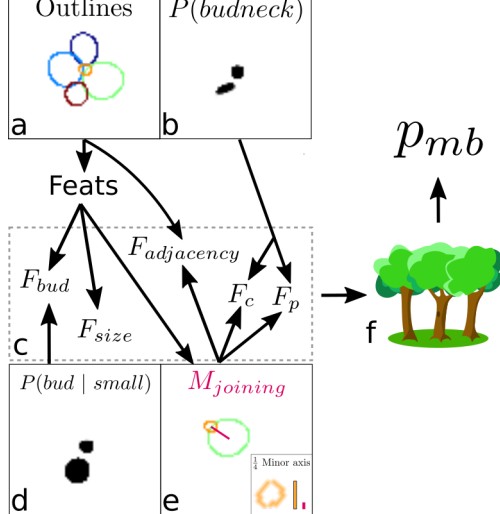

**Appendix 2—figure 4.** Overview of the algorithm for assigning lineages. (**a, b**) We start from the cell outlines and the CNN's predicted probabilities of a pixel being a bud neck for small cells. Different colour intensities show the
*Appendix 2—figure 4 continued on next page*

*Appendix 2—figure 4 continued*

probabilities with white denoting zero probability. (**c**) Composite features used by the classifier to solve the task. (**d**) The probability of small cells being a bud. (**e**) An intermediate element of assigning lineages is defining $M_{\text{joining}}$ – the red line, actually a rectangular box. (**f**) Feeding the features into a trained random forest model returns the probability of a pair of cells being a mother and bud.

We used multiple image features to characterise a mother-bud relationship. For an ordered pairing of all cells in an image, we consider a putative mother-bud pair and define a mask, $M_{\text{joining}}$, as the joining rectangle between the centres of the mother and bud with a width equal to one quarter the length of the bud's minor axis. Given $M_{\text{joining}}$, we consider five features:

i. $F_{\text{size}}$, which is the ratio of the mother's to bud's area. Mothers generally have a greater size than their bud so that $F_{\text{size}} > 1$.

ii. $F_{\text{adjacency}}$, which is the fraction of $M_{\text{joining}}$ intersecting with the union of the mother's and bud's masks. Mothers should be proximal to their buds so that $F_{\text{adjacency}}$ is close to one: only a small fraction of $M_{\text{joining}}$ should lie outside of the mother and bud outlines.

iii. $F_{\text{bud}}$, which is the mean over the union of the CNN's output for a small, interior targets and all pixels contained in the bud. $F_{\text{bud}}$ approximates the probability that a cell is a bud and should be close to one for mother-bud pairs.

iv. $F_{\text{p}}$, which is the mean over the union of the pixels contained in $M_{\text{joining}}$ with the CNN's output for bud-necks, only including those pixels whose probability is greater than 0.2. $F_{\text{p}}$ approximates the probability that a bud neck joins a mother and bud.

v. $F_{\text{c}}$, which is the number of the CNN's bud-neck target pixels with a probability greater than 0.2 that are in $M_{\text{joining}}$ normalised by the square root of the bud's area, or effectively the bud's perimeter. We interpret $F_{\text{c}}$ as a confidence score on $F_{\text{p}}$ because a single spurious pixel with high bud-neck probability could produce high $F_{\text{p}}$.

## Training a classifier

With these features, we train a random forest classifier to predict the probability that a pair of cells is a mother and bud. We train on all pairs of cells in the validation data. We optimised the hyperparameters, including the number of estimators and tree depth, using a grid search with five-fold cross-validation. We optimise for precision because true mother-bud pairs are in the minority and because our strategy for assigning lineages aggregates over multiple time points (as detailed below).

For our standard 5Z CNN trained to a pixel size of 0.182, the random forest classifier had a precision of 0.83 and recall of 0.54 on the test data. This precision and recall of the classifier to assign mother-bud pairs within a single time point is distinct from the precision and recall of mother-bud pairs after accumulating across time (reported in *Figure 3d*). This data has 211 true mother-bud pairs out of 1678 total pairs. The classifier assigned feature weights of 0.46 for $F_{\text{size}}$, 0.11 for $F_{\text{adjacency}}$, 0.24 for $F_{\text{bud}}$, 0.06 for $F_{\text{p}}$, and 0.13 for $F_{\text{c}}$.

## Assigning each cell a unique mother

To establish lineages, we need to assign at most one tracked mother cell to each tracked cell object. We use the classifier to assign a mother-bud probability $p_{mb}^{(t)}$ for each time point at which a pair of tracked objects are together. We then estimate the probability that a tracked object $\hat{c}$ has ever been a mother using

$$p_m^{(t)}\left(\hat{c}\right) = \max\left[p_m^{(t-1)}\left(\hat{c}\right), \sum_{c_b \in C_t} p_{mb}^{(t)}\left(\hat{c}, c_b\right)\right], \tag{10}$$

as well as the probability that it has ever been a bud with

$$p_b^{(t)}\left(\hat{c}\right) = \max\left[p_b^{(t-1)}\left(\hat{c}\right), \max_{c_m \in C_t}\left(p_{mb}^{(t)}\left(c_m, \hat{c}\right)\right)\right]. \tag{11}$$

Finally, we calculate a cumulative score for a putative mother-bud pair and reduce this score if the candidate bud has previously shown a high probability of being a mother:

$$S_{mb}^{(t)}\left(\hat{c}_m, \hat{c}_b\right) = S_{mb}^{(t-1)}\left(\hat{c}_m, \hat{c}_b\right) + p_{mb}^{(t)}\left(\hat{c}_m, \hat{c}_b\right)\left(1 - p_m^{(t)}\left(\hat{c}_b\right)\right). \tag{12}$$

At each time point, we then propose lineages by assigning each putative cell object $\hat{c}_b$ with a bud probability $p_b^{(t)}\left(\hat{c}_b\right) > 0.5$ and a mother $\hat{c}_m = \mathrm{argmax}_{\hat{c}}\left(S_{mb}^{(t)}\left(\hat{c}, \hat{c}_b\right)\right)$. We treat the mother-bud assignments proposed at the final time point as definitive because they have integrated information over the entire time series. To avoid spurious assignments, we require all buds to be present for at least three time points.

## Post-processing

Though rare, we do have to mitigate occasional detection, tracking and assignment errors. For example, debris can occasionally be mistakenly identified as a cell and tracked.

We discard tracks that have both small volumes and show limited growth over the experiment. Specifically, we discard a given cell track $i$ with duration $T_i$, minimal volume $V_i^{(\min)}$ at time $T_i^{(\min)}$, and maximal volume $V_i^{(\max)}$ at time $T_i^{(\max)}$, if both $V_i^{(\max)} < 7~\mu m^3$ and the estimated average growth rate $G_i < 10~\mu m^3/\mathrm{hour}$, where

$$G_i = \mathrm{sign}\left(T_i^{(\max)} - T_i^{(\min)}\right) \times \frac{V_i^{(\max)} - V_i^{(\min)}}{T_i}. \tag{13}$$

Our tracking algorithm usually identifies correctly instances where a mother and bud pivot with the flow of the medium, but exceptions do arise. For a given mother, we therefore join contiguous bud tracks – pairs of bud tracks where one ends with the other starting on the next time point – if the extrapolated volume of the old track falls within a threshold difference of the volume of the new track. Specifically, for the pair of contiguous tracks $i$ and $j$, with track $i$ ending at time point $t$ and track $j$ beginning at time point $t + 1$, we calculate

$$V_{ij}^{(\mathrm{diff})} = \min\left(\left|V_i^{(t)} + G_i \Delta T - V_j^{(t+1)}\right|, \left|V_j^{(t+1)} - G_j \Delta T - V_i^{(t)}\right|\right), \tag{14}$$

where $V_i^{(t)}$ is the volume of track $i$ at time point $t$ and $\Delta T$ is the time step between time points $t$ and $t + 1$. We join these tracks if $V_{ij}^{(\mathrm{diff})} < 7~\mu m^3$.

Finally, we discard any tracks with fewer than five time points.

## Appendix 3

### The BABY algorithm: estimating volumes and growth rates

## Calculating cell volumes

To estimate cell volumes, we model a 3D cell from our 2D outline. We use a conical method (*Gordon et al., 2007*), which is robust to common cell shapes, to estimate cell volume from an outline. This method makes two assumptions: that the outline obtained cuts through the widest part of the cell and that the cell is ellipsoidal. We build a cone with a base shape that is the filled outline of the cell by iteratively eroding the segmented mask of the cell and stacking these masks in the *Z* dimension. We find the volume of the cone by summing the voxels in the corresponding 3D mask. Finally, we multiply this sum by four to obtain the volume of the cell: a cone whose base is the equatorial plane of an ellipsoid will have a volume that is a quarter of the corresponding ellipsoid's volume (*Gordon et al., 2007*).

## Estimating single-cell growth rates

Depending on the need for computational speed, we use one of two methods for estimating instantaneous growth rates.

For long-term, and stored, analysis, we estimate growth rates by fitting a Gaussian process with a Matern covariance function to the time series of each cell's volume (*Swain et al., 2016*). We set the bounds on the hyperparameters to prevent over-fitting. Maximising the likelihood of the hyperparameters, we are able to obtain the mean and first and second time derivatives of the volume, as well as estimates of their errors. The volume's first derivative is the single-cell growth rate.

During real-time processing where we may wish to use to the growth rate to control the microscope, fitting a Gaussian process is too slow. Instead we estimate growth rates from the smoothed first derivative obtained by Savitzky-Golay filtering of each cell's volume time series. Though faster, this method is less reliable and does not estimate errors. For time series of mothers, we use a third-order polynomial with a smoothing window of seven time points; for time series of buds, we use a fourth-order polynomial also with a smoothing window of seven time points.

We estimate growth rates separately for mothers and their buds because both are informative. We find that the summed results are qualitatively similar to previous estimates of growth rate, which fit the time series of the combined volume of the mother and its bud (*Ferrezuelo et al., 2012*; *Cookson et al., 2010*).

## Appendix 4

## A graphical user interface for curating

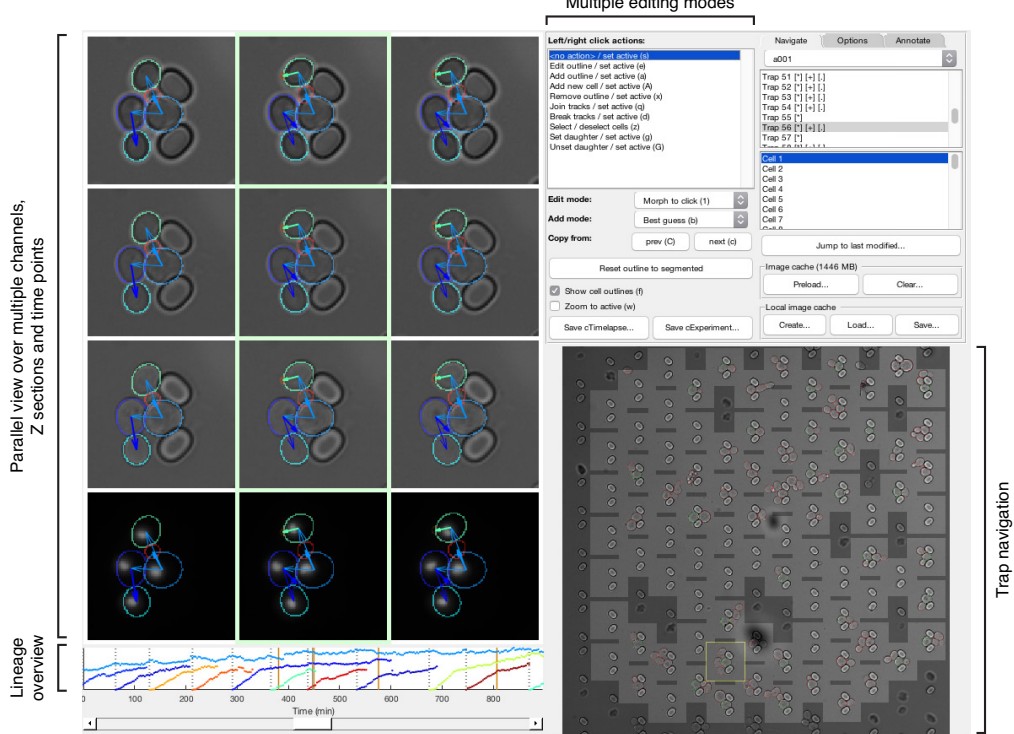

**Appendix 4—figure 1.** Main features of the graphical user interface used for annotation. We developed a custom graphical user interface (GUI) in Matlab to annotate efficiently overlapping cell instances, tracks and lineages over long time courses. The screen-shot shows the GUI in its horizontal layout with three bright-field sections and a fluorescence channel selected for parallel view. Annotated outlines and arrows indicating lineage relationships have each been toggled on for display. The GUI can display up to 9 time points in parallel; the slider at the bottom allows fast scrolling through the entire time-lapse. A time-course summary panel is displayed above the slider and has been set to show the outline areas for a mother and all its buds. An overview image of the entire position allows navigation between traps. The user can select from multiple editing modes for manipulating annotations in the parallel view region, including modes for draggable outline editing, track merging and splitting, and lineage reassignments.

To ease annotating overlapping instances, cell tracks and lineage relationships, we developed a Graphical User Interface (GUI) in Matlab that allows parallel viewing of multiple Z sections and time points (*Appendix 4—figure 1*). This parallel view helps curate buds obscured by a lack of focus and those that might be missed without simultaneously observing multiple time points. The GUI mirrors manipulations made to outlines and tracks to all views in real time. The interface is highly customisable, with multiple layouts available and the ability to select which sections and channels to display. To edit outlines for smaller cells, the user can adjust the level of zoom. Further, starting outlines can be copied across time points and interpolated forwards or backwards in time (interpolated outlines are annotated as such until they are manually adjusted).

The GUI saves annotations in a custom format for computational efficiency, but various export options are available. For training we exported annotations in PNG format with one image per time point. Because outlines can potentially overlap, they are tiled, with one cell instance per tile. We store track and lineage annotations in the metadata of the PNG file.

Furthermore, the GUI includes features to efficiently detect and correct rare errors. A track display panel provides visual aids to summarise tracks across the entire time course. In particular, the 'Display mother and daughter areas' mode uses this panel to plot the area of the currently selected cell and all of its daughters over the time course. Using this mode, many segmentation and tracking

errors are highlighted as unexpected jumps in area or changes in track label (denoted in colour). We use a slider to navigate to these errors where they can be either corrected in place or saved for future curation.

Although the GUI works with whole images, it includes features to navigate and annotate images that naturally partition into regions, such as the traps of our ALCATRAS devices. Then the trap navigation image shows trap locations and enables moving between traps.

## Appendix 5

### Quantifying localisation

During each experiment, we acquired bright-field and fluorescence images at five Z sections spaced 0.6 μm apart and used the maximum projection of these images (the maximum pixel values across all Z sections) for quantification.

For each cell, we determined its fluorescence image by subtracting the median fluorescence of the cell's pixels and setting all non-cell pixels to zero. A cytoplasmic pixel will determine this median fluorescence, and we assume that it results from autofluorescence only, which requires sufficiently low numbers of fluorescent markers.

To quantify fluorescent markers in the nucleus, we noted that fluorescence there appears in an image as a two-dimensional Gaussian distribution because of point spreading in epifluorescence microscopes. We therefore identified the most probable location of the nucleus for each cell by convolving a Gaussian filter with the fluorescence image. The maximal value in the resulting filtered image marks the location that most closely matches this filter.

Using data from nuclei segmented via Nhp6A-mCherry reporters (***Granados et al., 2018***), we observed that the area of the nucleus $A_{nuc}$ scales as a fraction of cell area $A_{cell}$ with a scaling factor $f_{nuc} \simeq 0.085$. We used this result to estimate a standard deviation σ for the Gaussian filter. If the nucleus is approximately circular then we estimate its radius as

$$r_{nuc} = \sqrt{\frac{f_{nuc}A_{cell}}{\pi}}. \tag{15}$$

Assuming that the segmented area of nucleus contains 95% of the nuclear fluorescence, we choose the σ of the Gaussian filter so that 95% of its probability is obtained by integrating over a circle of radius $r_{nuc}$. Writing $\alpha = 0.95$, we have

$$\begin{aligned} \alpha &= \int_{A_{nuc}} dxdy\, \frac{e^{-\frac{x^2+y^2}{2\sigma^2}}}{2\pi\sigma^2} \\ &= 2\pi \int_0^{r_{nuc}} dr\, r \frac{e^{-\frac{r^2}{2\sigma^2}}}{2\pi\sigma^2} \\ &= 1 - e^{-\frac{r_{nuc}^2}{2\sigma^2}} \end{aligned} \tag{16}$$

switching to polar coordinates. Using that the cumulative distribution function of the $\chi^2$ distribution with two degrees of freedom is $F_{\chi^2}(x) = 1 - e^{-x^2/2}$, we can rearrange ***Equation 16*** and combine with ***Equation 15*** to give

$$\sigma = \sqrt{\frac{f_{nuc}A_{cell}}{\pi F_{\chi^2}^{-1}(\alpha)}}. \tag{17}$$

We next assume an ideal fluorescence image of the nucleus can be described by the same Gaussian filter but re-scaled by some amplitude $a_f$. If we apply the Gaussian convolution $G$ to the pixel in this ideal image with maximal fluorescence, we obtain

$$a_f \left\| G^2 \right\| \tag{18}$$

where $\|G^2\|$ is the sum of the squared values of the Gaussian filter. This quantity should in principle be equal to $\alpha \max(C)$, where $C$ is the Gaussian filtered fluorescence image of the actual cell. Therefore

$$a_f = \frac{\alpha \max(C)}{\|G^2\|}. \tag{19}$$

Finally, $a_f$ is our prediction of the total nuclear fluorescence, but the concentration is more biologically relevant and, if denoted $N$, is

$$N = \frac{a_f}{f_{\mathrm{nuc}} A_{\mathrm{cell}}}$$
$$= \frac{\alpha \max(C)}{f_{\mathrm{nuc}} A_{\mathrm{cell}} \|G^2\|} \tag{20}$$

which is the measure we use.

For quantifying the localisation of Myo1-GFP to the bud neck, we note that $N$ is a sensitive proxy for localisation and assume that it applies equally well in this case.

## Appendix 6

### Estimating cytokinesis using fluorescent markers

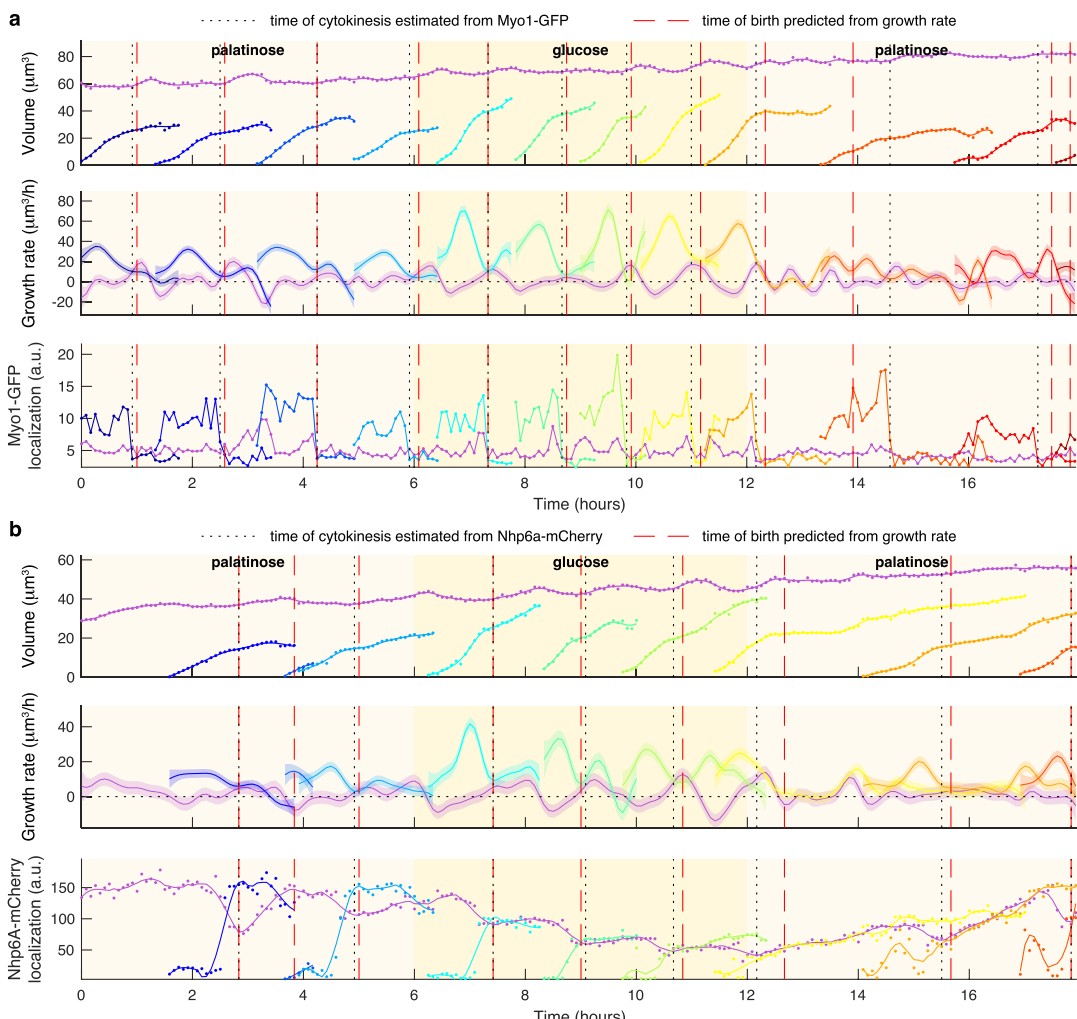

**Appendix 6—figure 1.** Markers for anaphase and cytokinesis reveal coincidence with a crossing point in mother and bud growth rates. (**a**) Time series for a mother (purple) and its buds and daughters for a switch from 2% palatinose to 2% glucose and back, with volumes and growth rates estimated by BABY. We use the localisation of Myo1-GFP to the bud neck to identify times of cytokinesis (vertical black dotted lines). For comparison, we show birth times predicted by our growth rate heuristic as vertical red dashed lines. (**b**) As for a, but using the localisation of Nhp6A-mCherry to the nucleus to identify times of cytokinesis (vertical black dotted lines). We show both the raw (points) and smoothed (lines; Savitzky-Golay filter with third degree polynomial and smoothing window of 15 time points) localisation of Nhp6A-mCherry.

We used either Myo1-GFP or Nhp6A-mCherry to estimate the time at which a bud becomes an independent daughter. Myo1, a type II myosin, localises to the bud neck and shows a drop in intensity upon cytokinesis (*Appendix 6—figure 1a*); Nhp6A, a histone-associated protein localised to the nucleus and shows a drop in intensity during anaphase as cells transport chromosomes into their buds (*Appendix 6—figure 1b*). Although anaphase and cytokinesis are distinct events in the cell cycle, the timing between the start of anaphase and completion of cytokinesis is similar across growth conditions (*Leitao and Kellogg, 2017*), and we assume cytokinesis occurs 20 min after anaphase. For *Figure 3—figure supplement 6a–d* and *Appendix 6—figure 1a*, we used Myo1-GFP; for *Figure 3—figure supplement 6e*, *Figure 4*, *Figure 4—figure supplement 1*, *Figure 5*, *Figure 5—figure supplement 1*, *Figure 6—figure supplement 1d*, and *Appendix 6—figure 1b*, we used Nhp6A-mCherry.

## Detecting cytokinesis from fluorescent Myo1

A drop in Myo1-GFP intensity at the mother cell's bud neck accompanies cytokinesis (*Figure 3—figure supplement 6a*), and we assume that this drop is fast compared to the time interval of imaging. We use the time series of fluorescence localisation within the mother cell over the period where the mother has a bud and estimate its time derivative using backward finite differences. To obtain candidate time points for cytokinesis, we find the dips in this derivative with a minimum prominence via Matlab's findpeaks function. We take the actual time point marking completion of cytokinesis as the last observed candidate peak before the next bud appears.

## Detecting anaphase from fluorescent Nhp6A

During anaphase, there is both a fall in fluorescence localisation of Nhp6A-mCherry in the mother and a rise in the bud (*Appendix 6—figure 1b*). Both signals typically level to a similar value as anaphase completes. For each bud, we therefore identify the start of anaphase using the difference between the mother and bud localisation signals, from when the bud appears to either when it disappears or the next bud appears. We set the start of anaphase as the last time point for which this mother-bud difference, normalised by its maximum, is greater than 0.5. We avoid selecting spurious differences after anaphase by considering only candidates that exist five time points before we observe a normalised difference under 0.1.

We ignore buds in further analysis for four reasons: we find no candidate time point for anaphase; the candidate is the first or second time point after the bud appears; the normalised difference does not drop below 0.1 within the 20 min following the candidate time, implying cytokinesis did not occur; the drop in the normalised localisation signal in the mother is less than or equal to 0.1.

## Predicting cytokinesis from growth rate

All together, we are able to determine key events of the cell cycle. First, we define a cell cycle for each mother as the duration between two budding events, obtained from the lineage assignment. These points approximately correspond to shortly after the START checkpoint (*Costanzo et al., 2004*). Second, assuming that the buds are accurately predicted, we identify a single point of cytokinesis within the corresponding cell cycle.

We observe three phases of growth during a cell cycle (*Figure 3—figure supplement 6a–b*). First, the bud dominates growth during S/G2/M, with its growth rate peaking midway through that period while simultaneously the mother's growth rate falls. Second, the bud's growth rate decreases as cytokinesis approaches. Near cytokinesis, the mother's and bud's growth rates have similar magnitudes, becoming identical at multiple time points. Finally, the mother's growth rate increases after cytokinesis, peaking during G1.

Observing that cytokinesis typically occurs where the peak in bud growth rate ends, we developed an algorithm to estimate the point of cytokinesis. For each bud, we consider its growth rate from its appearance to either its disappearance, the first appearance of its own bud, or the appearance of its mother's next bud. The time point of cytokinesis is then identified as the first time point after the maximum growth rate for which either the growth rate drops below zero or the derivative of the growth rate, estimated as the second derivative of the Gaussian process fit to the volume, rises above a threshold $g$. If neither condition holds, we set cytokinesis to the last of the time points considered.

We determined the threshold $g$ from a training set of 150 ground-truth estimates of cytokinesis determined by the Nhp6A-mCherry marker (30 from each condition in *Figure 4*). We evaluated accuracy using ground-truth estimates of cytokinesis determined from either the Myo1-GFP or Nhp6A-mCherry markers, excluding those used in training (*Figure 3—figure supplement 6*). Across multiple conditions, our method predicts the timing of cytokinesis to within two time points (6–10 min) for over 60% of the examples. A potential issue, however, is that we can compare only with cells for which we are able to assign at least one valid cell cycle using Myo1 or Nhp6A. There are multiple predictions made by the growth-rate method that we therefore ignore because there is no corresponding ground truth, and discarding these predictions may affect the overall result.

We used this method to predict cytokinesis for *Figure 1e*, *Figure 3—figure supplement 6* and *Appendix 6—figure 1*.

## Appendix 7

### Correlating nuclear Sfp1 with growth rate

The cross correlation of time series can reveal regulatory relationships (*Dunlop et al., 2008*). We applied cross correlations to investigate if fluctuations in Sfp1's localisation anticipate fluctuations in growth rate. Analysis by the method of *Kiviet et al., 2014* assumes steady-state cells. We nonetheless make use of data with a switch from palatinose to glucose and back (*Figure 4b*), but limit ourselves to time points from either the four hours preceding the switch to glucose – approximately steady growth in palatinose – or the four hours preceding the switch back to palatinose – approximately steady growth in glucose.

Correlations may occur on scales longer than the duration of a cell cycle, so we analysed only mother cells present over the full four hours of steady growth. We used the summed mother and bud growth rates whenever a bud is present because most of the mother's growth is in the bud. We identified when a daughter separates from its mother using Nhp6A-mCherry (Appendix 6). The medium washes away almost all daughters before they become mothers, making the lineage trees in our data have no branches and simplifying the analysis.

For each mother $i$, we have a time series $\ell_1^{(i)}, \ldots, \ell_N^{(i)}$ of the degree of localisation of Sfp1-GFP and a time series of instantaneous growth rates $g_1^{(i)}, \ldots, g_N^{(i)}$. For our sampling interval of $\Delta t = 5$ min, $N$ is 48. We denote the total number of mother cells by $M$ and calculate the deviation from the population mean for each time series:

$$\delta\ell_t^{(i)} = \ell_t^{(i)} - \frac{1}{M}\sum_j \ell_t^{(j)} \quad \text{and} \quad \delta g_t^{(i)} = g_t^{(i)} - \frac{1}{M}\sum_j g_t^{(j)}. \tag{21}$$

The cross-covariance of Sfp1 localisation and growth rate at a time lag of $r\Delta t$ is then (*Kiviet et al., 2014*):

$$C_{lg}^{(i)}(r\Delta t) = \begin{cases} \frac{1}{N-r}\sum_{t=1}^{N-r} \delta\ell_t^{(i)} \cdot \delta g_{t+r}^{(i)} & \text{if } r \geq 0 \\ C_{gl}^{(i)}(-r\Delta t) & \text{otherwise.} \end{cases} \tag{22}$$

We find the cross-correlation through normalising by the standard deviations:

$$R_{lg}^{(i)}(r\Delta t) = \frac{C_{lg}^{(i)}(r\Delta t)}{\sqrt{C_{\ell\ell}^{(i)}(0)C_{gg}^{(i)}(0)}}. \tag{23}$$

We determined the auto-correlation for Sfp1 localisation, $R_{\ell\ell}^{(i)}(r\Delta t)$, and for growth rate, $R_{gg}^{(i)}(r\Delta t)$, similarly. In *Figure 5c–d* of the main text, we show the mean and 95% confidence interval over all mother cells (all $i$).

## Appendix 8

## Real-time feedback control

In these experiments, we wished to trigger a change in media based on the cells' growth rate. As an example, we switched medium from a richer to a poorer carbon source and used BABY to determine how long we should keep cells in this medium for approximately 50% to have resumed dividing before we switch back to the richer medium.

We ran code to implement the feedback control on two computers: one controlling the microscope (Appendix 8 Algorithm 2) and the other both segmenting images, via calls to Python, and determining growth rates (Appendix 8 Algorithm 3). The code is in Matlab and available on request.

We defined the fraction of escaped cells as the proportion of included mothers that have had a bud or daughter exceed a threshold in growth rate of 15 $\mu m^3$ /hr at any time point after the onset of the lag in growth caused by the poorer carbon source. We defined this lag period to begin at the time point when the median daughter growth rate first drops below 5 $\mu m^3$ /hr. To be included, a mother cell must satisfy two requirements: be present in our data for at least 95% of the time points from the 20 time points before the first switch to the current time point; and have an assigned bud or daughter for at least 10% of the time we observe it.

To increase processing speed, we used Savitzky-Golay filtering to estimate growth rates. The resulting first derivative is not well constrained at the end-points, making instantaneous growth rates vary widely at the most recently acquired time point. We therefore used growth rates up to and including the time point three steps before the most recent when determining the fraction of escaped cells.

We used the strain BY4741 Sfp1-GFP Nhp6A-mCherry in both experiments.

---

Algorithm 2 **Feedback control – pseudocode for microscope acquisition software**

---

```
Set glucose pump to infuse at 4µl/min;
Set ethanol pump off;
for 270 timepoints do
    image acquired time = current time + 5 min Acquire images at 6 stage positions
    Save images in networked directory
    while current time<image acquired time do
        if time since start ≥ 5hours and first switch has not happened then
            Run switch protocol (fast infuse/withdraw to remove back pressure);
            Set glucose pump off;
            Set ethanol pump to infuse at 4µl/min;
        end
        read onlinedata.txt
        if fraction of escaped mothers is recorded and second switch has not happened then
            if fraction of escaped mothers ≥ 0.5 then
                Run switch protocol (fast infuse/withdraw to remove back pressure);
                Set glucose pump to infuse at 4µl/min;
                Set ethanol pump off;
            end
        end
    end
end
```

---

Algorithm 3 **Feedback control – pseudocode for segmentation software**

```
for 270 timepoints do
    while segmentation of all positions is not complete do
        for 6 positions do
            Check networked data directory
            if all images for current position are recorded then
                Run BABY segmentation on current position
            end
        end
    end
    for 6 positions do
        Calculate growth rates by Savitzky-Golay filtering
        Append result to array for all positions
    end
    Write median growth rate to onlinedata.txt
    if first switch has happened then
        Calculate lag start time as first time point where median growth rate <5µl/hour
        if lag start has happened then
            Calculate fraction of escaped mothers and write to onlinedata.txt
        end
    end
end
```

