## [Editor Report]

The authors develop important machine-learning approaches to extract single-cell growth rates and show convincing evidence that their methods can yield insight into growth control. They also introduce compelling new methodologies for several other aspects of automated image analysis.

---

## [Decision Letter]

**Decision letter after peer review:**

Thank you for submitting your article "A label-free method to track individuals and lineages of budding cells" for consideration by *eLife*. Your article has been reviewed by 3 peer reviewers, one of whom is a member of our Board of Reviewing Editors, and the evaluation has been overseen by Naama Barkai as the Senior Editor. The reviewers have opted to remain anonymous.

The reviewers have discussed their reviews with one another, and the Reviewing Editor has drafted this to help you prepare a revised submission. Overall, all the reviewers were impressed by the innovative techniques and potential biological insights in the paper. At the same time, however, after discussion, the reviewers were not convinced that the work represents a "method" in the sense that it could actually be used by others or on data from other labs. Further, the reviewers all had major problems with the clarity of the manuscript, finding it difficult to fully understand either the key methodological or biological contributions.

Essential revisions:

1) Convince the reader that the "method" described here can be applied in other settings and that improved identification of small buds will be a generally important advance towards estimating growth rates from movies. As it stands, the reviewers wondered if the contribution here was really only a solution to a specific data analysis problem that arises in the specific imaging/microfluidics set-up considered here. Ideally, this issue could be addressed by clearly specifying the problem(s) that the method is designed to solve (e.g., is it bud identification, cell tracking and/or growth rate estimation), and clear comparison of this method to state-of-the-art methods on a variety of datasets (with appropriate held out test data and statistics.)

2) Clarify what the major novelty/contributions/discoveries are in the main manuscript text, describe them fully, and move all the other clever innovations (that are not fully supported by empirical data) to the supplementary materials or Methods section. For each contribution, give the appropriate context (previous work, hypothesis, importance, etc.) and ensure that the generality of the claims match the experiments/data presented (e.g., generality of observation of sizer vs timer regulating bud growth or generality of the tracking results beyond cells grown in traps). Ensure that the Title and Abstract reflect what has actually been demonstrated in the paper.

3) Overall, clarify the writing to give the appropriate scientific context for a general biology audience, and remove or explain technical jargon that will not be familiar to the readers of *eLife*.

*Reviewer #1 (Recommendations for the authors):*

Specific issues:

– I am left wondering if this a software that other groups can expect to download and use, or is it an abstract "method" to make the point that it's possible to accurately segment small buds and extract growth rates and other labs should implement similar systems for their own imaging setups?

– Quite a bit of discussion of "morphological erosion" which I could not really follow. Is it the same usage as this?

https://en.wikipedia.org/wiki/Erosion_(morphology)

The authors should define what they mean by this term and explain why it is important.

– Very hard for me to understand the training and testing data: what is the data used? Was the method ever evaluated on data from other microscopes/labs/benchmarks? The only numbers I got were the training images 588 trap images – 1813 annotated cells in total.

– Sfp1 experiments are explained relatively clearly compared to the rest of the paper, but still major improvements are needed. E.g., it is never stated that the authors are assuming nuclear localization of Sfp1 reflects its "activity" (what is even meant, transcriptional activity?) While plausible, what is the evidence that this is true? Similarly, "enters the nucleus in response to two conserved nutrient-sensing kinases" – again this is not specific enough. How does Sfp1 "respond"? Presumably the kinases phosphorylate something and eventually Sfp1 translocates into the nucleus, and the authors are quantifying the GFP-signal in the nucleus? Non- experts will not be able to follow this without much more specific explanation.

– "With BABY, we add the ability – with no extra imaging costs – to measure what is often our best estimate of fitness, single-cell growth rates." one of the points of the paper is that the cell size/growth information can be obtained 'for free' (i.e., with no additional labels) from microfluidics experiments. Have the authors actually gone back and reanalyzed any (even of their own) previously published data to illustrate this point? As it stands it seems a bit like a rhetorical argument.

– "To better characterise how growth rate varies during the cell cycle, we ran experiments for cells with the gene MYO1 tagged with Green Fluorescent Protein" Very hard to follow this. "Ran experiments" is not scientific. What is the logic? What was measured? "We determined that we could predict cytokinesis from growth rate because at cytokinesis the mother's and bud's growth rates often reach a similar magnitude (Appendix 5)." I can't follow this. What is the evidence that cytokinesis can be predicted? How is this measured? What are the alternative models that were rejected? Do I need to read Appendix 5? I found this claim in Appendix 5: "We evaluated the algorithm by the correlation between the real time of cytokinesis and the predicted time" What were the sample sizes for the test data? etc., etc., How was the "real" time measured? Is it this: "The actual time of cytokinesis is taken to be the candidate point with the minimal derivative, corresponding to the strongest down-shift in fluorescence."?

*Reviewer #2 (Recommendations for the authors):*

1. It is not entirely clear to me how successful the automated bud-mother assignment actually is. In the main text (page 6) the authors talk about 80% precision. However, in the appendix we learn that the recall is only 47%. What is the actual accuracy of correct mother-bud identifications after the downstream assignment? How many tracks are rejected by the post-processing? The accuracy of the final pedigree analysis needs to be reported more directly.

2. Along those lines, the results shown in Figure 3c and d make it very hard to assess the actual accuracy of the automated extraction of cell cycle information. Especially the low number of only 6 daughter cells in Figure 3c is far from sufficient, as can be also seen by the unusual bimodal distribution – which does not reflect the distributions in Figure 3d. Instead, the authors should compare the results in Figure 3d to a manual analysis of the same cells (or at least a significant subfraction). Rather than histograms, it would be informative to see plots showing predicted vs manually analyzed TM and TD or a distribution of differences between those values in manually annotated and predicted data.

3. Unfortunately, the correlations shown in Figure 2a and c of Appendix 5 are mostly meaningless. The high correlation is simply a consequence of the fact that absolute time of the experiment is shown, and obviously cells born later in the experiment will finish cytokinesis later (or have a later time of predicted cytokinesis). As can be seen from this plot, predictions often deviate dramatically (on the timescale of hours!) from the ground truth. We need to see the same plot using time relative to bud emergence rather than absolute time.

4. The plots shown in Figure 3a and b are fine, but it would be beneficial to also see more standard metrics, such as average precision vs IoU threshold (see for example https://doi.org/10.1038/s41592-020-01018-x), or MOTA for tracking.

5. To allow fair comparison, segmentation by BABY using a single z-slice should

be compared with e.g. YeaZ (and ideally other deep learning methods) retrained with the same data.

6. It is not entirely clear to me whether Figure 4d and e rely on the Myo1-GFP signal or are purely obtained with automated BABY analysis. The Myo1-based data are a nice opportunity to further quantify the accuracy of the BABY-based cell cycle predictions. Again it would be nice to see predicted vs Myo1-based cell-cycle duration plotted against each other.

7. The biological results shown in Figure 4 are very interesting. However, I think to support the claim of sizer-timer based bud growth regulation, more than two conditions are needed. Does this claim still hold with multiple different media, or even appropriate mutants?

8. If I understand correctly, buds by definition are defined as independent cells if they become larger than 24 fL (page 33)? How many buds are affected by this criteria? How does this match with Figure 4e, where we clearly see bigger buds?

9. The claim in Figure S1 that YeaZ underestimates bud size more than BABY should be supported by a more direct plot, e.g. showing YeaZ (and BABY) predictions vs ground truth bud size on a single bud level.

10. Is the choice of evaluating 10 sets of hyperparameters in Appendix 3, Figure 3 determined by limiting computational resources? If not, this seems rather sparse, especially given the fact that panel b does not give the impression that a parameter set close to the optimum is found. Would testing more combinations not likely result in a significant improvement?

*Reviewer #3 (Recommendations for the authors):*

I would recommend the authors rewrite the paper focusing on the very interesting biology. Then, they can choose whatever method they like to analyze their data since that is not the main focus of the paper and there are no claims as to which method is better than which other.

---

## [Author Response]

Essential revisions:1) Convince the reader that the "method" described here can be applied in other settings and that improved identification of small buds will be a generally important advance towards estimating growth rates from movies. As it stands, the reviewers wondered if the contribution here was really only a solution to a specific data analysis problem that arises in the specific imaging/microfluidics set-up considered here. Ideally, this issue could be addressed by clearly specifying the problem(s) that the method is designed to solve (e.g., is it bud identification, cell tracking and/or growth rate estimation), and clear comparison of this method to state-of-the-art methods on a variety of datasets (with appropriate held out test data and statistics.)

We now emphasise that BABY’s main innovation is estimating single-cell growth rates. Its ability to do so is by identifying buds both accurately and early through segmenting overlapping cells. That ability comes from defining multiple targets of differing cell sizes for the neural network. We expect this solution to be widely applicable and have changed the paper’s title accordingly.

To demonstrate the generality of our approach, we have performed extensive analyses, comparing BABY not only with YeaZ but also with Cellpose, a state-of-the-art, deep-learning based segmentation algorithm. Figure 3 is completely new as a consequence, and we have added a new section on evaluating BABY’s performance in Methods.

We show that BABY generalises, accurately analysing YeaZ’s microcolony training data (Figure 3, figure supplement 2). Although this training data is labelled, it does not allow for overlaps. We therefore manually annotated three of its 45 fields-of-view and added these to the BABY training data. We then assessed performance on the remaining 42 fields-of-view using the annotations from the YeaZ paper. Because YeaZ’s curation lacks overlaps, BABY has no particular advantage, but it nonetheless shows competitive performance with YeaZ and the Cellpose algorithm when we train all on the same data. We further note that some of the false positives detected by BABY were, upon inspection of the images, true positive buds, which were not annotated in the YeaZ training data, perhaps because they overlapped substantially with other cells.

For our own curated images with traps, we now compare BABY with both the Cellpose and YeaZ algorithms, showing results after retraining on the same training data as BABY (Figure 3a and 3b). For fair comparison, we focus on BABY’s performance when the input is only a single Z section. We nonetheless continue to highlight the advantage of using multiple Z sections (Figure 3a, 3c and Figure 3 supplement 1d-f), and we use this highest-performing model in our applications.

BABY produces the lowest errors in growth rates (Figure 3c), mostly because it more accurately segments cells with small sizes (Figure 3b).

We also now demonstrate the importance of detecting buds when estimating growth rates with a new supplement to figure 1 (Figure 1 figure supplement 1). This figure shows that smaller buds have the fastest growth rates, particularly in the size range where we observe overlaps (Appendix 1, Figure 2). The data are from a new benchmark data set, which we manually segmented specifically for testing the accuracy of estimating growth rates.

2) Clarify what the major novelty/contributions/discoveries are in the main manuscript text, describe them fully, and move all the other clever innovations (that are not fully supported by empirical data) to the supplementary materials or Methods section. For each contribution, give the appropriate context (previous work, hypothesis, importance, etc.) and ensure that the generality of the claims match the experiments/data presented (e.g., generality of observation of sizer vs timer regulating bud growth or generality of the tracking results beyond cells grown in traps). Ensure that the Title and Abstract reflect what has actually been demonstrated in the paper.

We now highlight BABY’s innovations, particularly the choice of targets for the U-net, which bring most of the improved performance and can be used widely in other image-processing applications. We have shortened the manuscript, removing the section on identifying cytokinesis and reducing the discussion. As well as the changes shown in blue, we have made minor edits throughout to improve focus and clarity.

We have changed the title and removed and re-written parts of the abstract.

3) Overall, clarify the writing to give the appropriate scientific context for a general biology audience, and remove or explain technical jargon that will not be familiar to the readers of eLife.

Throughout we are now careful both to define technical terms used in image processing and machine learning and to minimise their use.

Reviewer #1 (Recommendations for the authors):Specific issues:– I am left wondering if this a software that other groups can expect to download and use, or is it an abstract "method" to make the point that it's possible to accurately segment small buds and extract growth rates and other labs should implement similar systems for their own imaging setups?

Our paper aims to provide both of these things. Researchers can freely download and use BABY on their data, but we believe too that others will incorporate its novel techniques into their own image-processing software.

As discussed in Essential revisions, we now demonstrate that BABY works not only on images with traps but also on microcolonies. We also show the importance of accurately segmenting small buds when estimating growth rates.

– Quite a bit of discussion of "morphological erosion" which I could not really follow. Is it the same usage as this?https://en.wikipedia.org/wiki/Erosion_(morphology)The authors should define what they mean by this term and explain why it is important.

We have minimised and now explain image-processing jargon.

– Very hard for me to understand the training and testing data: what is the data used? Was the method ever evaluated on data from other microscopes/labs/benchmarks? The only numbers I got were the training images 588 trap images – 1813 annotated cells in total.

We have moved where we describe the training data from Appendix 2 to a new “Training data” subsection in Methods and provide more detail on the sources of the images. We also now demonstrate that BABY works on microcolony data from another laboratory, using data from YeaZ.

– Sfp1 experiments are explained relatively clearly compared to the rest of the paper, but still major improvements are needed. E.g., it is never stated that the authors are assuming nuclear localization of Sfp1 reflects its "activity" (what is even meant, transcriptional activity?) While plausible, what is the evidence that this is true? Similarly, "enters the nucleus in response to two conserved nutrient-sensing kinases" – again this is not specific enough. How does Sfp1 "respond"? Presumably the kinases phosphorylate something and eventually Sfp1 translocates into the nucleus, and the authors are quantifying the GFP-signal in the nucleus? Non- experts will not be able to follow this without much more specific explanation.

We have re-written this section. Sfp1 is directly phosphorylated by TORC1 and likely too by PKA and moves into the nucleus because of these phosphorylations. We no longer say “activity”, only “nuclear localisation”.

– "With BABY, we add the ability – with no extra imaging costs – to measure what is often our best estimate of fitness, single-cell growth rates." one of the points of the paper is that the cell size/growth information can be obtained 'for free' (i.e., with no additional labels) from microfluidics experiments. Have the authors actually gone back and reanalyzed any (even of their own) previously published data to illustrate this point? As it stands it seems a bit like a rhetorical argument.

We have changed “no extra imaging costs” to “using only bright-field images”, which we hope the reviewer agrees is clearer.

– "To better characterise how growth rate varies during the cell cycle, we ran experiments for cells with the gene MYO1 tagged with Green Fluorescent Protein" Very hard to follow this. "Ran experiments" is not scientific. What is the logic? What was measured? "We determined that we could predict cytokinesis from growth rate because at cytokinesis the mother's and bud's growth rates often reach a similar magnitude (Appendix 5)." I can't follow this. What is the evidence that cytokinesis can be predicted? How is this measured? What are the alternative models that were rejected? Do I need to read Appendix 5? I found this claim in Appendix 5: "We evaluated the algorithm by the correlation between the real time of cytokinesis and the predicted time" What were the sample sizes for the test data? etc., etc., How was the "real" time measured? Is it this: "The actual time of cytokinesis is taken to be the candidate point with the minimal derivative, corresponding to the strongest down-shift in fluorescence."?

We agree that this section was too terse and its importance over emphasised, and we no longer include the method described in the main BABY algorithm. We have therefore moved the section to Appendix 6, where we have improved its clarity, expanding on how we define cytokinesis.

Reviewer #2 (Recommendations for the authors):1. It is not entirely clear to me how successful the automated bud-mother assignment actually is. In the main text (page 6) the authors talk about 80% precision. However, in the appendix we learn that the recall is only 47%. What is the actual accuracy of correct mother-bud identifications after the downstream assignment? How many tracks are rejected by the post-processing? The accuracy of the final pedigree analysis needs to be reported more directly.

We apologise for the confusion. Those precision and recall values were for assignments given a single time point. Our algorithm, however, benefits by aggregating over time, and we now evaluate both precision and recall for pairing mother and bud tracks (Figure 3d). We used our tracking test set, but supplemented with extra manually curated tracks to cover an additional two growth conditions.

The tracks we reject with post-processing are those with small volumes, low growth rates, or with durations under five time points. If such tracks correspond to cells, they are almost always not the central cells – those caught in the centre of the microfluidic traps – because these cells are usually stably trapped. As we now show in Figure 3d, BABY’s recall reduces if we assess performance against all tracks rather than those for central cells. We do not, however, consider this decrease important for most applications. In our laboratory at least, the focus has always been on the central cells because of their stability. With up to a thousand in one video, this restriction causes no shortage of data.

2. Along those lines, the results shown in Figure 3c and d make it very hard to assess the actual accuracy of the automated extraction of cell cycle information. Especially the low number of only 6 daughter cells in Figure 3c is far from sufficient, as can be also seen by the unusual bimodal distribution – which does not reflect the distributions in Figure 3d. Instead, the authors should compare the results in Figure 3d to a manual analysis of the same cells (or at least a significant subfraction). Rather than histograms, it would be informative to see plots showing predicted vs manually analyzed TM and TD or a distribution of differences between those values in manually annotated and predicted data.

We agree with the reviewer that the performance on extracting cell-cycle information was ambiguous. These results partly relied on our prediction of points of cytokinesis from growth rates. We now consider the accuracy of these predictions insufficient and on longer include predicting cytokinesis in the main BABY algorithm. Consequently, we have removed the results showing the extracted cell-cycle information entirely.

3. Unfortunately, the correlations shown in Figure 2a and c of Appendix 5 are mostly meaningless. The high correlation is simply a consequence of the fact that absolute time of the experiment is shown, and obviously cells born later in the experiment will finish cytokinesis later (or have a later time of predicted cytokinesis). As can be seen from this plot, predictions often deviate dramatically (on the timescale of hours!) from the ground truth. We need to see the same plot using time relative to bud emergence rather than absolute time.

Thank you: this error was one we should have caught.

We now compare with the predicted time of cytokinesis from the most recent budding event (Appendix 6). Although we predict cytokinesis within two time points (ten minutes) of the ground truth for 60% of the cells tested, we feel that this accuracy is too low compared to the accuracy of the rest of BABY, and we have removed the cytokinesis predictions from the main algorithm.

4. The plots shown in Figure 3a and b are fine, but it would be beneficial to also see more standard metrics, such as average precision vs IoU threshold (see for example https://doi.org/10.1038/s41592-020-01018-x), or MOTA for tracking.

We now report the mean average precision versus IoU threshold for all tested algorithms in Figure 3a.

We also evaluated tracking using the Multiple Object Tracking Performance metric (Figure 3 supplement 5), but found that it did not distinguish well between the tested methods. We suspect that this result is because the MOTA metric aims to assess cases where many objects are tracked with frequent errors. In our setting, however, we believe it is more important to have as many high precision tracks of the correct length rather than correctly identifying many short-lived tracks. We therefore include an alternative, more intuitive tracking metric in Figure 3c and have moved our previous track-overlap metric to Figure 3 supplement 4.

5. To allow fair comparison, segmentation by BABY using a single z-slice shouldbe compared with e.g. YeaZ (and ideally other deep learning methods) retrained with the same data.

In Figure 3, we now highlight and focus on the performance of the single Z-slice model. We also compare with models trained on our data, both for YeaZ and Cellpose.

6. It is not entirely clear to me whether Figure 4d and e rely on the Myo1-GFP signal or are purely obtained with automated BABY analysis. The Myo1-based data are a nice opportunity to further quantify the accuracy of the BABY-based cell cycle predictions. Again it would be nice to see predicted vs Myo1-based cell-cycle duration plotted against each other.

Those panels relied on using fluorescent Nhp6A, a nuclear marker, to estimate when anaphase starts. We have altered Figure 4’s legend and axes labels to clarify this use of Nhp6A and highlight the time at which we expect cytokinesis, approximately 20 minutes after anaphase begins.

To avoid confusion with our prediction of cytokinesis from growth rate – now dropped from the main BABY algorithm, we have moved the original panels Figure 4a-c to a figure in Appendix 6.

7. The biological results shown in Figure 4 are very interesting. However, I think to support the claim of sizer-timer based bud growth regulation, more than two conditions are needed. Does this claim still hold with multiple different media, or even appropriate mutants?

We have supplemented our analysis with data from three additional growth conditions and more correctly identify the previously reported glucose growth condition as following a transition from growth in palatinose.

8. If I understand correctly, buds by definition are defined as independent cells if they become larger than 24 fL (page 33)? How many buds are affected by this criteria? How does this match with Figure 4e, where we clearly see bigger buds?

We exclude a track from being assigned as a bud if the volume of the corresponding cell was always above 24 fL. Objects above this size are often indistinguishable from small cells that the flow of medium transiently washes into the field-of-view from upstream in the microfluidic device. We have added this information to Appendix 2.

If BABY identifies small buds early enough and they are correctly tracked, then the size of a bud may grow beyond 24 fL.

9. The claim in Figure S1 that YeaZ underestimates bud size more than BABY should be supported by a more direct plot, e.g. showing YeaZ (and BABY) predictions vs ground truth bud size on a single bud level.

We have removed this figure. Figure 3e now systematically compares BABY and YeaZ, and Cellpose, to the ground truth.

10. Is the choice of evaluating 10 sets of hyperparameters in Appendix 3, Figure 3 determined by limiting computational resources? If not, this seems rather sparse, especially given the fact that panel b does not give the impression that a parameter set close to the optimum is found. Would testing more combinations not likely result in a significant improvement?

We have removed this panel, which showed the performance of U-nets with differing numbers of filters and depths. As the reviewer implies, it is not particularly informative. Nonetheless, we consistently found that U-nets with fewer parameters than the standard one, which has a depth of d = 5 and n = 64 filters, showed similar, if not slightly higher, performance, but use less memory and storage and are faster.

Reviewer #3 (Recommendations for the authors):I would recommend the authors rewrite the paper focusing on the very interesting biology. Then, they can choose whatever method they like to analyze their data since that is not the main focus of the paper and there are no claims as to which method is better than which other.

Here unfortunately we have to disagree with the reviewer. We believe that a methods paper will have higher impact if it has accompanying examples illustrating the new insights the method brings.